# Presynaptic Rac1 in the hippocampus selectively regulates working memory

Jaebin Kim[1], Edwin Bustamante[1], Peter Sotonyi[2], Nicholas Maxwell[1], Pooja Parameswaran[1], Julie K Kent[1], William C Wetsel[1,3,4,5], Erik J Soderblom[1,6], Bence Rácz[7], Scott H Soderling[1,5]*

[1]Department of Cell Biology, Duke University School of Medicine, Durham, United States; [2]Department of Anatomy and Histology, University of Veterinary Medicine, Budapest, Hungary; [3]Department of Psychiatry and Behavioral Sciences, Duke University School of Medicine, Durham, United States; [4]Mouse Behavioral and Neuroendocrine Analysis Core Facility, Duke University School of Medicine, Durham, United States; [5]Department of Neurobiology, Duke University School of Medicine, Durham, United States; [6]Proteomics and Metabolomics Shared Resource and Center for Genomic and Computational Biology, Duke University School of Medicine, Durham, United States; [7]Department of Anatomy and Histology, University of Veterinary Medicine, Budapest, Hungary

**\*For correspondence:**
scott.soderling@duke.edu

**Competing interest:** The authors declare that no competing interests exist.

**Abstract** One of the most extensively studied members of the Ras superfamily of small GTPases, Rac1 is an intracellular signal transducer that remodels actin and phosphorylation signaling networks. Previous studies have shown that Rac1-mediated signaling is associated with hippocampal-dependent working memory and longer-term forms of learning and memory and that Rac1 can modulate forms of both pre- and postsynaptic plasticity. How these different cognitive functions and forms of plasticity mediated by Rac1 are linked, however, is unclear. Here, we show that spatial working memory in mice is selectively impaired following the expression of a genetically encoded Rac1 inhibitor at presynaptic terminals, while longer-term cognitive processes are affected by Rac1 inhibition at postsynaptic sites. To investigate the regulatory mechanisms of this presynaptic process, we leveraged new advances in mass spectrometry to identify the proteomic and post-translational landscape of presynaptic Rac1 signaling. We identified serine/threonine kinases and phosphorylated cytoskeletal signaling and synaptic vesicle proteins enriched with active Rac1. The phosphorylated sites in these proteins are at positions likely to have regulatory effects on synaptic vesicles. Consistent with this, we also report changes in the distribution and morphology of synaptic vesicles and in postsynaptic ultrastructure following presynaptic Rac1 inhibition. Overall, this study reveals a previously unrecognized presynaptic role of Rac1 signaling in cognitive processes and provides insights into its potential regulatory mechanisms.

## eLife assessment

The paper characterized a specific defect in the spatial working memory of mice with a deficit in a protein called Rac1. Rac1 inhibition was limited to the presynaptic compartment of neurons, which is significant because past work has inhibited both pre- and postsynaptic compartments. The study also identified potential effectors of Rac1. The work is **important** for these reasons, and the strength of the evidence is **exceptional**.

## Introduction

Ras-related C3 botulinum toxin substrate 1 (Rac1) is a small Rho GTPase that functions as a molecular switch cycling between active GTP-bound and inactive GDP-bound states in response to extracellular signals. Rac1 participates in a variety of cellular functions, including gene expression, endocytosis, and cytoskeletal remodeling, which are facilitated through interactions with a wide variety of effector proteins, including kinases and actin regulatory factors (*Bosco et al., 2009*; *Déléris et al., 2011*; *Etienne-Manneville and Hall, 2002*; *Tejada-Simon, 2015*). In neurons, Rac1 is important for its role in actin cytoskeleton organization, a process fundamental for neuronal migration, morphogenesis, and synaptic plasticity (*Gordon-Weeks and Fournier, 2014*; *Kawauchi et al., 2003*; *Kunschmann et al., 2019*; *Spence and Soderling, 2015*). Its significance is particularly evident in postsynaptic compartments, where Rac1 and its downstream effector kinases and actin regulatory WAVE1 complex are known to influence spine morphology and neurotransmission. For instance, Rac1 activation in mouse cortical and hippocampal neurons increases spine density while reducing spine size (*Tashiro et al., 2000*). Additionally, Rac1 is critical for maintaining the structural long-term plasticity of spines (*Hedrick et al., 2016*; *Tu et al., 2020*), and the Rac1/WAVE1/Arp2/3 pathway is involved in the actin remodeling that regulates dendritic spine formation (*Soderling et al., 2007*). These effects are linked to signaling pathways downstream of synaptic receptor activation that are known to be important for learning and memory, via Guanine nucleotide Exchange Factors (GEFs). For example, the activation of NMDA receptors has been shown to trigger a cascade leading to the phosphorylation of kalirin-7, a GEF, in a process dependent on CaMKII activation. This sequence of events is vital for spine structural and functional plasticity (*Xie et al., 2007*). Tiam1, another Rac-GEF, is also critical for multiple synapse functions, including spinogenesis, plasticity, and learning and memory (*Cheng et al., 2021*; *Kojima et al., 2019*; *Li et al., 2023*; *Rao et al., 2019*; *Saneyoshi et al., 2019*). Moreover, the diffusion of Rac1 signaling, facilitated by BDNF–TrkB activation, is essential for the structural enhancement of nearby spines, contributing to their long-term potentiation (*Hedrick et al., 2016*).

Consistent with Rac1's important roles in synaptic signaling, mutations of Rac1 or alterations to its signaling pathways are associated with a variety of neurological disorders (*Fatemi et al., 2013*; *Hayashi-Takagi et al., 2010*; *Huang et al., 2020*). Dysregulation of Rac1 leads to abnormalities in synaptogenesis (*Wiens et al., 2005*), spine morphology (*Nakayama et al., 2000*), and synaptic plasticity (*Haditsch et al., 2009*), each of which has been proposed to be a contributing factor for various developmental brain disorders, including autism spectrum disorder (ASD) and schizophrenia. For example, altered Rac1 activity is a factor in ASD-like behaviors in the DOCK4-deficient mouse model of ASD (*Guo et al., 2021*), and a mutation in the switch II region of RAC1 is implicated in neurodevelopmental disorders (*Banka et al., 2022*). Additionally, mutations in the Rac-GEF Trio are associated with autism and intellectual ability and impair neuroligin-1 induced synaptogenesis (*Fingleton and Roche, 2023*; *Tian et al., 2021*).

Rac1 is expressed broadly in brain regions; however, its function in the hippocampus has been most extensively studied. In the hippocampus, the Rac1-cofilin signaling regulates F-actin dynamics and thereby affects both long-term potentiation and depression, key processes in modulating synaptic strength and thereby influencing learning and memory (*Kim et al., 2009*; *Tu et al., 2020*; *Wang et al., 2023*). Additionally, ablation of Rac1 in postmitotic neurons significantly impairs hippocampus-dependent working memory, with subtle effects on behaviors linked to longer-term forms of memory (*Haditsch et al., 2009*). While actin cytoskeleton remodeling is believed to be a key contributor to Rac1's involvement in learning and memory, the specific molecular mechanisms translating actin remodeling to altered synaptic efficacy remain to be explored.

Although the majority of the above literature has focused on the postsynaptic spine, actin remodeling also governs several aspects of presynaptic function (*Nelson et al., 2013*; *Rust and Maritzen, 2015*). Yet comparatively less is known about the signaling events linked to presynaptic actin. Recently, we showed that presynaptic Rac1 is activated by action potentials and negatively regulates the replenishment rate of synaptic vesicles of hippocampal neurons and thus modulates short-term presynaptic plasticity (*O'Neil et al., 2021*). This finding underscores the importance of understanding presynaptic Rac1 in answering the long-standing question of how presynaptic signaling and short-term presynaptic plasticity may influence learning and memory.

In this study, we explore the behavioral impacts and regulatory mechanisms of Rac1 at presynaptic terminals. We demonstrate that mice exhibit an impairment in spatial working memory when

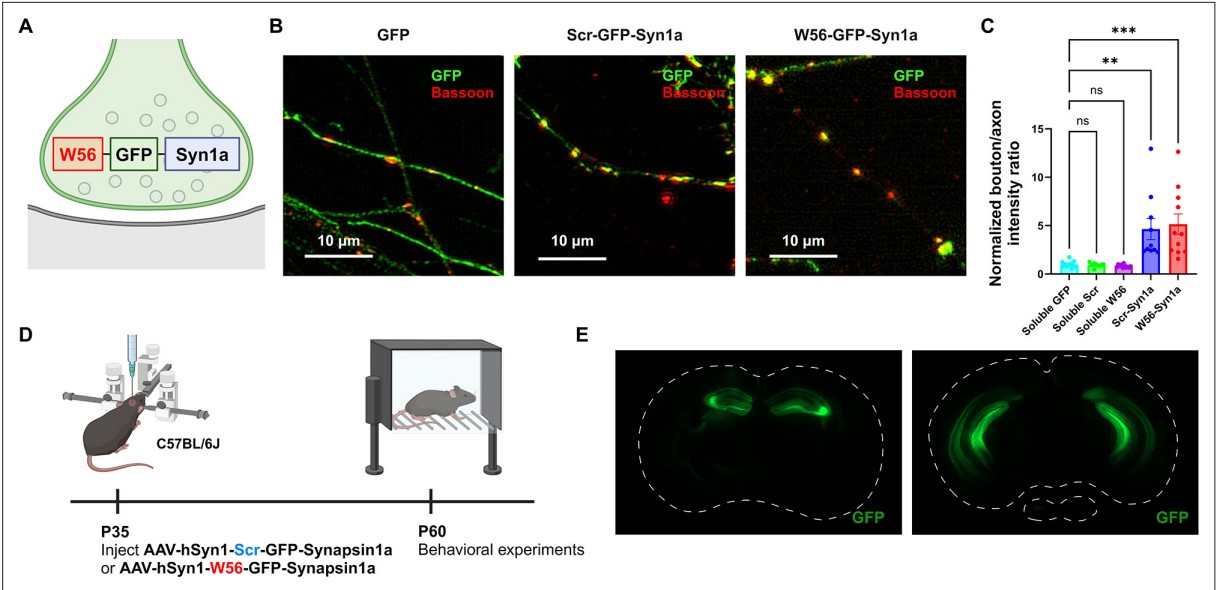

**Figure 1.** Localization of the presynaptic Rac1 inhibitor construct. (**A**) Illustration of the presynaptic Rac1 inhibitor construct. (**B**) C57BL/6J primary neuron cultures expressing GFP, the negative control construct, or the presynaptic Rac1 inhibitor construct. (**C**) Fluorescence intensity profiles validated presynaptic localization of the constructs (Soluble GFP: $n = 11$ cells; Soluble Scr: $n = 8$ cells; Soluble W56: $n = 11$ cells; Scr-Syn1a: $n = 10$ cells; W56-Syn1a: $n = 11$ cells). (**D**) Timeline of AAV injection and behavioral experiments. (**E**) Representative immunohistochemistry images of brain slices expressing W56-GFP-Syn1a. Dotted lines indicate the outlines of the brain slices. Data are expressed as mean ± standard error of the mean (SEM) with ns. not significant, $**p < 0.01$ and $***p < 0.001$.

Rac1 inhibitors are expressed at presynaptic terminals and in longer-term memory when expressed at postsynaptic sites, suggesting a site-specific involvement of Rac1 in learning and memory processes. Importantly, our findings suggest that the presynaptic functions of Rac1 are intricately mediated through the phosphorylation of synaptic vesicle proteins. This phosphoregulation, coupled with actin cytoskeleton remodeling, may play a pivotal role in modulating the synaptic vesicle cycle and plasticity, underscoring a sophisticated mechanism by which Rac1 may influence synaptic dynamics and, consequently, cognitive processes.

## Results

### Inhibition of Rac1 activity at presynaptic terminals in the hippocampus impairs spatial working memory

To explore the behavioral effects of presynaptic Rac1, we assessed the hippocampal-dependent learning and memory in C57BL/6J mice with presynaptic Rac1 inhibition. We inhibited Rac1 activity specifically at presynaptic terminals using the presynaptic Rac1 inhibitor construct (*O'Neil et al., 2021*), which is composed of a Rac1-inhibiting polypeptide W56 (*Gao et al., 2001*) fused to a presynaptic protein Synapsin1a (Syn1a) (*Figure 1A*). W56 was replaced with a scrambled sequence (Scr) in the negative control group. Previous studies have demonstrated that W56 effectively inhibits Rac1 activity (*Gao et al., 2001*), and this inhibitory effect can be spatially targeted when W56 is fused to a protein with specific subcellular localizations in neurons (*Hedrick et al., 2016*). Our previous work confirmed that targeting W56 to presynaptic terminals could recapitulate the presynaptic phenotypes of Cre-mediated knockout of Rac1 as well as optical inhibition of Rac1 signaling, demonstrating its effectiveness (*O'Neil et al., 2021*). We verified the presynaptic localization of the presynaptic Rac1 inhibitor construct in primary neuron cultures (*Figure 1B, C*). We stereotaxically injected adeno-associated viruses (AAVs) encoding this construct bilaterally throughout the hippocampus of C57BL/6J mice and conducted behavioral experiments following a recovery period (*Figure 1D*). We also verified the expression of the constructs in the dorsal and ventral hippocampus (*Figure 1E*).

Using mice expressing either W56 or the scrambled peptide fused to Synapsin1a in the hippocampus, we next performed behavioral analysis to probe potential physiological functions of presynaptic Rac1. In the radial-arm maze test, mice were allowed to freely explore the eight-arm maze for 5 min (*Figure 2A*). Post hoc analysis showed that the average number of arm entries and latency to the first arm entry were not significantly different between the control group and the W56 group (*Figure 2B, C*), suggesting that ambulatory activity and anxiety levels were not affected by presynaptic Rac1 inhibition. However, mice in the W56 group exhibited impairment in spatial working memory as they made the first error, or entered a previously visited arm, in fewer entries and in less time relative to the control (*Figure 2D, E*).

Because the radial-arm maze test requires a relatively high memory load, we also conducted the spontaneous alternation Y-maze test, in which mice were allowed to freely explore the three-arm maze for 5 min (*Figure 2F*). Unexpectedly, the W56 group exhibited higher ambulatory activity as the number of arm entries was greater relative to the control (*Figure 2G*). However, anxiety levels are unlikely to account for this difference as the latency to the first arm entry was similar (*Figure 2H*). To avoid hyperactivity affecting measurement of working memory performance, we calculated the percentage of spontaneous alternations, which is the number of consecutive entries into each of the three arms divided by the maximum possible number of alternations (# of entries minus two). The percentage of spontaneous alternations of the W56 group was significantly lower than in the control group and closer to the chance level (*Figure 2I*). The total time spent in the decision area, or the center of the maze, was unchanged (*Figure 2J*).

To more comprehensively assess memory performance beyond basic exploratory behaviors and spatial recognition in mice, we conducted the delayed non-matching to position (DNMTP) T-maze test, which included habituation, pre-training, and DNMTP task (*Figure 2K*). During habituation, mice were handled for 1 min each day. During pre-training, we performed six forced trials daily, where each mouse was placed on the maze with one goal arm blocked and the food reward in the opposite, open goal arm. The DNMTP task involved six pairs of a forced trial and a choice trial with an inter-trial interval (ITI) of 5 s. In the choice trial, a food reward was placed in the goal arm opposite to the one that was open in the preceding forced trial. The mouse was then placed at the start arm, with both goal arms open. The percentage of correct choices of the control group gradually increased over time and plateaued approximately at 80%, which was consistent with previous studies (*Kilonzo et al., 2021*; *Wietrzych et al., 2005*). In contrast, the W56 group exhibited slower learning and fewer correct choices in the last 3 days of the test (*Figure 2L*). Overall, these data demonstrate that presynaptic Rac1 inhibition in the hippocampus impairs spatial working memory performance in mice. We did not observe any significant difference in the light/dark transition and the open field tests (*Figure 2M–U*), suggesting that working memory performance was not affected by anxiety or locomotor activity.

## Inhibition of Rac1 activity at presynaptic terminals does not affect other types of memory

Rac1 is involved in spatial learning and other types of memory, including long-term memory (LTM) (*Liu et al., 2016*). To assess whether presynaptic Rac1 inhibition is sufficient to affect cognitive functions beyond working memory, we performed the Morris water maze (MWM), a well-established test for hippocampal-dependent learning and memory (*Morris et al., 1982*; *Logue et al., 1997*; *Haditsch et al., 2009*). The task involved two sessions of acquisition trials with an ITI of 30 min daily for 6 consecutive days, followed by a probe trial 30 min after the last acquisition trial on days 2, 4, and 6 (*Figure 3A*). Swimming speed during acquisition trials was consistent for both groups throughout the test (*Figure 3B*). Escape latency of the control and W56 groups decreased over days at similar rates (*Figure 3C*), implying that spatial learning was intact in the W56 group. Furthermore, the percentage of time spent in the quadrant from which the platform was removed during the probe trials was also not significantly different (*Figure 3D*), suggesting that spatial reference memory and the rate of memory consolidation were not affected by presynaptic Rac1 inhibition.

One report suggests that pharmacological inhibition of Rac1 activity in the hippocampus may alter forgetting in fear conditioning, indicating Rac1's potential role in associative learning and emotional memory (*Jiang et al., 2016*). To test whether presynaptic Rac1 is involved in this process, we performed a fear conditioning experiment (*Figure 3E, F*). In the training session, each mouse was placed in a chamber for 3 min, during which a 30-s auditory cue followed by an electric foot shock

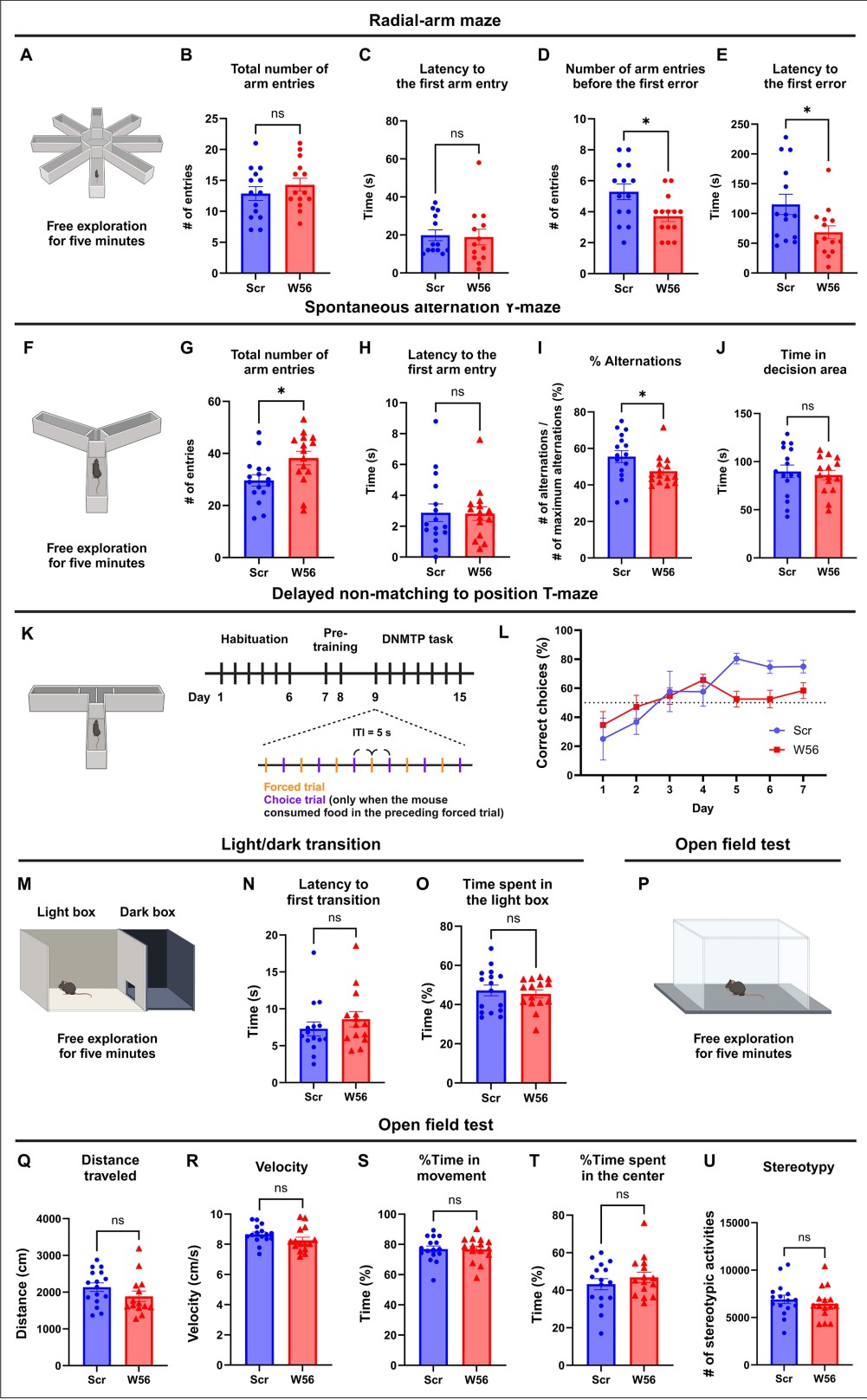

**Figure 2.** Hippocampal presynaptic Rac1 inhibition impairs spatial working memory. (**A**) Schematic of the eight-arm radial maze test (*n* = 14 mice per group). (**B**) The total number of arm entries (Scr: 12.86 ± 1.123, W56: 14.29 ± 1.051, p = 0.3615) and (**C**) latency to the first arm entry (Scr: 23.93 ± 4.877 s, W56: 26.07 ± 8.106 s, p = 0.5020) were not significantly different, whereas (**D**) the number of arm entries before the first error (Scr: 5.286 ± 0.5074,

*Figure 2 continued on next page*

*Figure 2 continued*

W56: 3.714 ± 0.3544, p = 0.0174) and (**E**) latency to the first error (Scr: 115.3 ± 17.11 s, W56: 68.36 ± 11.01 s, p = 0.0248) were lower in the W56 group. (**F**) Schematic of the spontaneous alternation Y-maze test (*n* = 15 mice per group). (**G**) The total number of entries (Scr: 29.63 ± 2.185, W56: 38.20 ± 2.599, p = 0.0168) was higher in the W56 group, while (**H**) latency to the first arm entry (Scr: 2.875 ± 0.5719 s, W56: 2.811 ± 0.4422 s, p = 0.7039) was similar. (**I**) The percentage of alternations (Scr: 55.58 ± 3.34%, W56: 47.51 ± 2.12%, p = 0.030) was significantly lower in the W56 group. (**J**) Time spent in the decision area (Scr: 89.85 ± 6.555 s, W56: 86.03 ± 4.873 s, p = 0.6467) was not influenced by presynaptic Rac1 inhibition. (**K**) Schematic of the delayed non-matching to position (DNMTP) T-maze test (*n* = 8 mice per group). (**L**) Percentage of correct choice trials during the DNMTP task. (**M**) Schematic of the light/dark transition test (*n* = 16 mice per group). (**N**) The latency to the first transition (Scr: 8.263 ± 1.313 s, W56: 9.827 ± 1.571, p = 0.2130) and (**O**) percentage of time spent in the light box (Scr: 47.22 ± 2.750%, W56: 45.42 ± 2.008%, p = 0.6054) were not significantly different. (**P**) Schematic of the open field test (Scr: *n* = 16 mice; W56: *n* = 15 mice). No significant change was observed in (**Q**) the distance traveled (Scr: 2135 ± 117.6 cm, W56: 2030 ± 195.9 cm, p = 0.1417), (**R**) velocity (Scr: 8.665 ± 0.1518 cm/s, W56: 8.262 ± 0.2067 cm/s, p = 0.1242), (**S**) percentage of time in movement (Scr: 76.91 ± 2.035%, W56: 76.67 ± 2.103%, p = 0.9347), (**T**) percentage of time spent in the center (Scr: 43.20 ± 2.980%, W56: 46.84 ± 2.822%, p = 0.3842), and (**U**) number of stereotypic activities (Scr: 6912 ± 445.5, W56: 6470 ± 430.4, p = 0.4827). Data are expressed as mean ± standard error of the mean (SEM) with ns, not significant, and *p < 0.05.

was applied. Contextual fear conditioning sessions were conducted at 1 hr, 24 hr, and 10 days post-training by placing each mouse back in the same chamber for 5 min. A cued fear conditioning session was performed 2 days post-training by placing the mice in a novel spatial context and exposing them to the same auditory cue approximately halfway through the session. The percentage of time spent freezing was measured and averaged in 1 min bins, except for 30 s bins during and immediately after the auditory cue and electric shock. Mice with presynaptic Rac1 inhibition displayed freezing for similar durations as the negative control in all sessions (*Figure 3G*), suggesting that retrieval of fear memory was unaffected by presynaptic Rac1 inhibition.

## Inhibition of Rac1 activity at postsynaptic sites affects remote memory

To assess whether Rac1 is involved in learning and memory in a site-specific manner, we examined the effects of postsynaptic Rac1 inhibition in C57BL/6J mice. We developed two postsynaptic Rac1 inhibitor constructs by fusing W56 with a linker to fibronectin intrabodies generated with mRNA display (FingRs) targeting a postsynaptic marker, either PSD95 or Gephyrin (*Bensussen et al., 2020*; *Gross et al., 2013*), with the scrambled sequence replacing W56 in the negative control group (*Figure 4A*). PSD95 and gephyrin FingRs recognize each endogenous protein and have been used extensively to localize fusion proteins such as fluorescent proteins to postsynaptic sites (*Gross et al., 2016*; *Kim et al., 2023*; *Rimbault et al., 2024*; *Trimmer, 2022*). We combined AAVs encoding these postsynaptic Rac1 inhibitor constructs in equal ratios and injected the mixture bilaterally throughout the hippocampus of adult C57BL/6J mice (*n* = 16 each for control and W56 group). We confirmed the expression of the postsynaptic Rac1 inhibitor constructs at postsynaptic sites (*Figure 4B, C*, *Figure 4—figure supplement 1A*) and in the hippocampus (*Figure 4—figure supplement 1B*). Behavioral tests, including the light/dark transition and the open field tests, revealed no significant effects of postsynaptic Rac1 inhibition on anxiety or locomotor activity (*Figure 4—figure supplement 2A–G*).

In the radial-arm maze and spontaneous alternation Y-maze tests, there was no notable change in spatial working memory (*Figure 4D–I*). In the DNMTP T-maze test, both W56 and control groups showed a similar gradual increase in the percentage of correct choices over time (*Figure 4J*). In the MWM, escape latency for both groups decreased at comparable rates, and no significant difference was found in time spent in the platform quadrant (*Figure 4K, L*), indicating intact LTM in mice with postsynaptic Rac inhibition. In the fear conditioning test, the percentage of freezing time was not significantly different overall, with the exception of the contextual trial 10 days after training (*Figure 4M*). In this trial, the W56 group exhibited a significantly reduced level of freezing, suggesting an impairment in remote fear memory. This difference was not likely driven by shock sensitivity because the response to shock was not different between treatment groups (*Figure 4—figure supplement 2H*). Overall, the behavioral experiments suggested that postsynaptic Rac1 inhibition does not influence

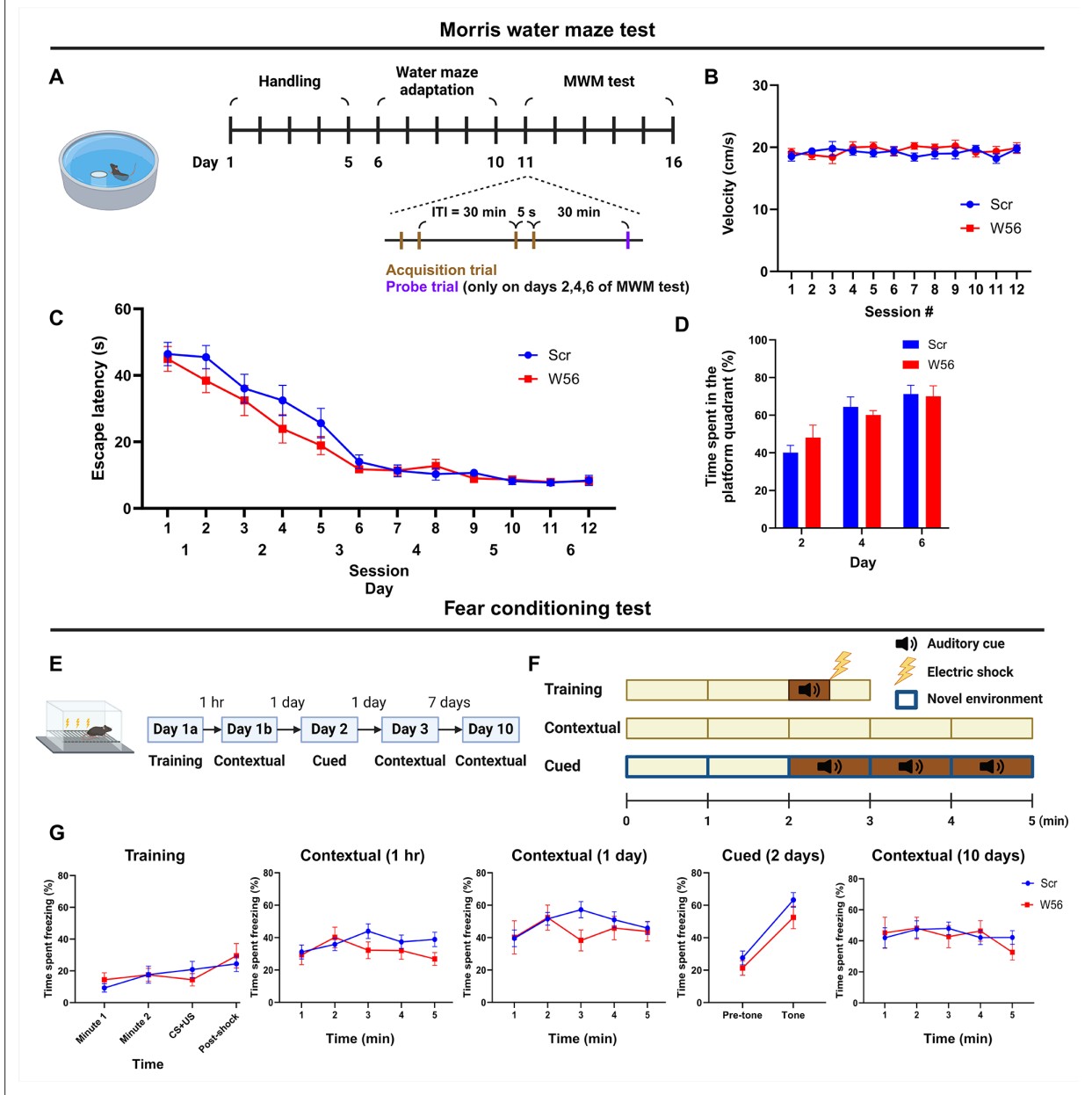

**Figure 3.** Hippocampal presynaptic Rac1 inhibition does not affect other types of learning and memory. (**A**) Schematic and timeline of the Morris water maze (MWM) test (Scr: $n$ = 16 mice; W56: $n$ = 15 mice). During the MWM acquisition trials, (**B**) average swimming speed and (**C**) escape latency showed no significant differences across treatment groups (two-way repeated measures analysis of variance [ANOVA]; Swimming speed: $F_{(1,29)}$ = 0.253, p = 0.619; Escape latency: $F_{(1,29)}$ = 1.40, p = 0.246). (**D**) Percentage of time spent in the platform quadrant during the probe trials was also not significantly different across treatment groups ($F_{(1,29)}$ = 0.024, p = 0.878). (**E**) Timeline of the fear conditioning test (Scr: $n$ = 16 mice; W56: $n$ = 15 mice). (**F**) Schematic of fear conditioning sessions. (**G**) Percentage of time spent freezing. Data are expressed as mean ± standard error of the mean (SEM).

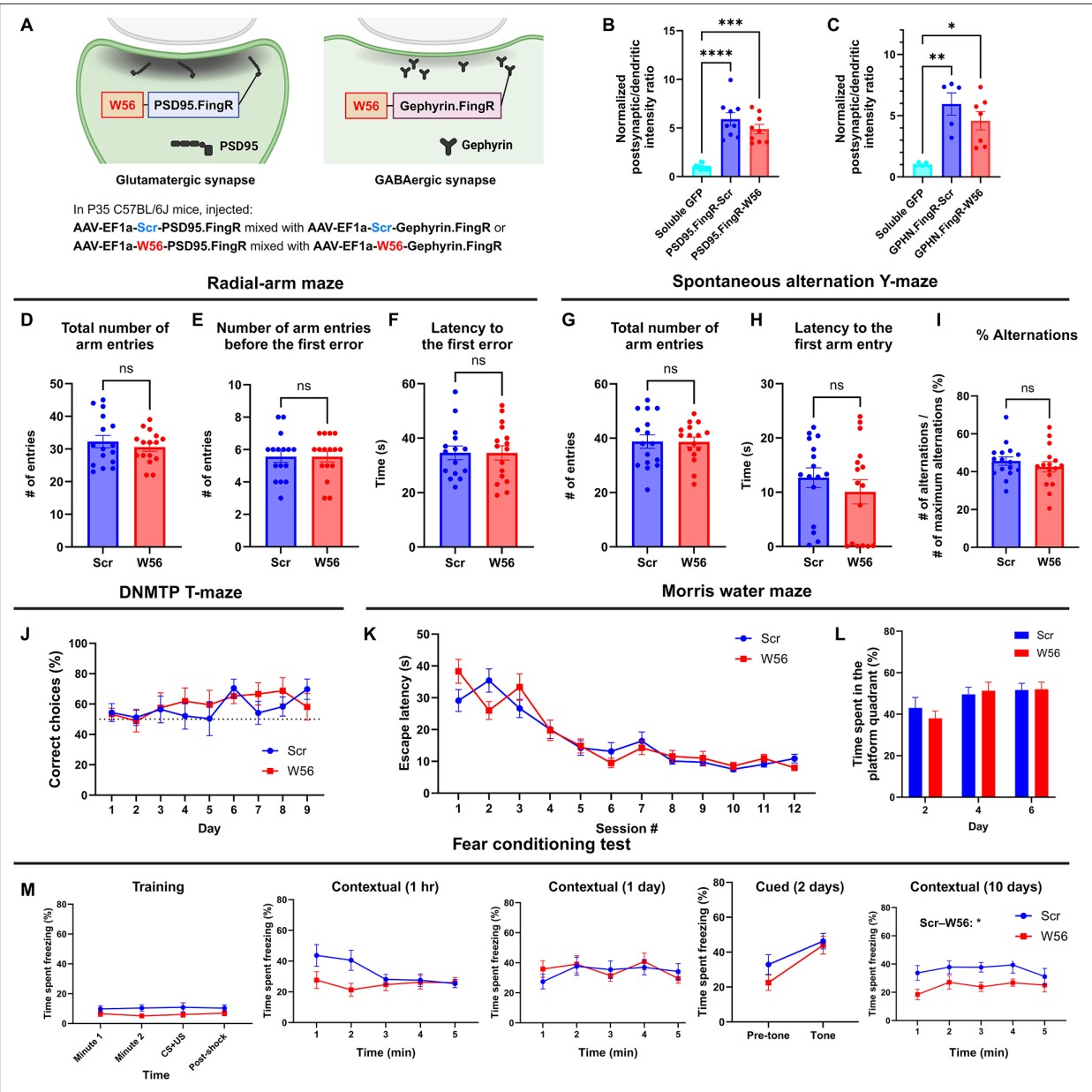

**Figure 4.** Hippocampal postsynaptic Rac1 inhibition does not affect hippocampal-dependent learning and memory. (**A**) Illustration of the postsynaptic Rac1 inhibitor constructs. (**B, C**) Fluorescence intensity profiles validated postsynaptic localization of the constructs (Soluble GFP (Homer): $n = 12$ cells; PSD95.FingR-Scr: $n = 9$ cells; PSD95.FingR-W56 : $n = 9$ cells; Soluble GFP (Gephyrin): $n = 5$ cells; GPHN.FingR-Scr: $n = 5$ cells; GPHN.FingR-W56: $n = 7$ cells). The radial-arm maze ($n = 16$ mice per group) revealed no significant differences in (**D**) the total number of arm entries (Scr: 32.25 ± 1.896, W56: 30.56 ± 1.245, $p = 0.4627$), (**E**) number of arm entries before making the first error (Scr: 5.563 ± 0.3532, W56: 5.563 ± 0.3287, $p = 0.8199$), and (**F**) latency to the first error (Scr: 37.19 ± 3.500 s, W56: 34.56 ± 2.630 s, $p = 0.9919$). The spontaneous alternation Y-maze ($n = 16$ mice per group) indicated no significant difference in (**G**) the total number of arm entries (Scr: 38.75 ± 2.452, W56: 38.63 ± 1.798, $p = 0.9675$), (**H**) latency to the first arm entry (Scr: 12.68 ± 1.819 s, W56: 10.07 ± 2.240 s, $p = 0.3226$), and (**I**) percentage of alternations (Scr: 45.61 ± 2.254%, W56: 42.56 ± 2.723%, $p = 0.3953$). (**J**) The percentage of correct choice trials in the DNMTP T-maze ($n = 8$ mice per group) was comparable for both groups throughout the test. In the Morris water maze (Scr: $n = 16$ mice; W56: $n = 15$ mice), (**K**) the escape latency from acquisition sessions ($F_{(1,29)} = 0.04019$, $p = 0.8425$), and (**L**) the time spent in the platform quadrant ($F_{(1,29)} = 0.05692$, $p = 0.8131$) also revealed no significant differences. (**M**) Percentage of time spent freezing in the fear conditioning test (Scr: $n = 15$ mice; W56: $n = 14$ mice). The difference between treatment groups was significant during the contextual trial 10 days after training (two-way repeated measures analysis of variance [ANOVA]; Treatment: $F_{(1,27)} = 6.169$, $p = 0.0195$), but not in other trials. Data are expressed as mean ± standard error of the mean (SEM) with ns, not significant, *$p < 0.05$, **$p < 0.01$, ***$p < 0.001$, and ****$p < 0.0001$.

*Figure 4 continued on next page*

*Figure 4 continued*

The online version of this article includes the following figure supplement(s) for figure 4:

**Figure supplement 1.** Localization of postsynaptic Rac1 inhibitor constructs.

**Figure supplement 2.** Anxiety and general locomotor activity are not affected by postsynaptic Rac1 inhibition.

hippocampal-dependent spatial working memory but does affect forms of longer-term memory such as remote fear memory.

## Postmitotic presynaptic Rac1 inhibition affects synaptic morphology, vesicle size, and distribution

Manipulation of Rac1 activity influences synaptic morphogenesis, likely via actin cytoskeleton regulation (*Costa et al., 2020*; *Zamboni et al., 2018*). While the role of Rac1 in postsynaptic morphology is established, its impact on presynaptic terminals is still unclear. To test whether presynaptic Rac1 inhibition induces any synaptic structural change, we analyzed hippocampal neurons with presynaptic Rac1

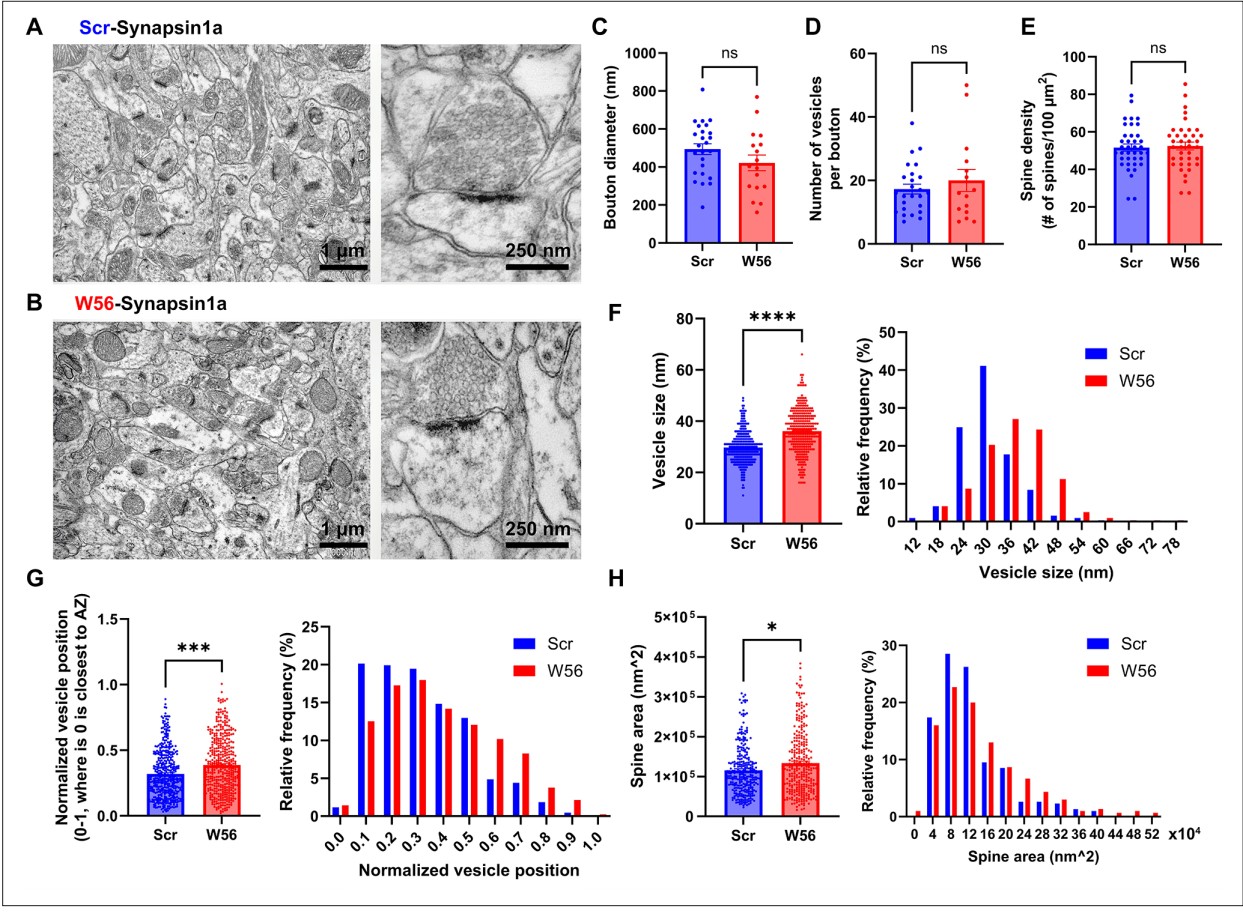

**Figure 5.** EM analysis of presynaptic Rac1 inhibition in hippocampal cells. Representative electron microscopy (EM) images of the dorsal CA1 stratum radiatum transduced with AAV-hSyn1-Scr-Syn1a (**A**) and AAV-hSyn1-W56-Syn1a (**B**). (**C**) Diameter of axonal boutons (Scr: 493.8 ± 28.38 nm, $n$ = 25 boutons; W56: 421.5 ± 41.47 nm, $n$ = 17 boutons; p = 0.1437). (**D**) Number of synaptic vesicles per bouton (Scr: 17.28 ± 1.552, $n$ = 25 boutons; W56: 20.00 ± 3.486, $n$ = 17 boutons; p = 0.9877). (**E**) Number of dendritic spines per 100 µm² (Scr: 51.64 ± 1.927 spines/100 µm²; W56: 52.46 ± 2.068 spines/100 µm²; $n$ = 38 images per group; p = 0.7730). (**F**) Bar graph and frequency distribution of diameter of synaptic vesicles show that presynaptic Rac1 inhibition leads to larger vesicles (Scr: 30.02 ± 0.3845 nm, W56: 36.35 ± 0.4993 nm; $n$ = 321 vesicles per group; p < 0.0001). (**G**) Synaptic vesicles were located relatively further from the active zone in the W56 group (Scr: 0.3193 ± 0.008885, $n$ = 432 vesicles; W56: 0.3869 ± 0.01032, $n$ = 423 vesicles; p ≈ 0.0001). The position of individual vesicles within a presynaptic terminal was normalized as a value ranging from 0 to 1, where 0 is closest to the active zone. (**H**) Area of dendritic spines was larger following presynaptic Rac1 inhibition (Scr: 124,400 ± 4342 nm², $n$ = 305 spines; W56: 144,700 ± 5676 nm², $n$ = 300 spines; p = 0.0276). Kolmogorov–Smirnov test was performed on data that did not pass normality test. Data are expressed as mean ± standard error of the mean (SEM) with ns, not significant, *p < 0.05, ***p < 0.001, and ****p < 0.0001.

inhibition using electron microscopy (EM). The presynaptic Rac1 inhibitor construct W56-Synapsin1a or the negative control Scr-Synapsin1a was expressed throughout the hippocampus through AAV transduction in adult P60 C57BL/6J mice (*n* = 3 per group). After 4 weeks, we fixed brain tissues and examined the dorsal CA1 stratum radiatum neuropil using EM (*Figure 5A, B*).

We observed no significant differences in bouton diameter, number of synaptic vesicles per bouton, and spine density between the two groups (*Figure 5C–E*). However, the W56 group exhibited a notable increase in diameter of vesicle (*Figure 5F*), implying possible disruptions in synaptic vesicle endocytosis by presynaptic Rac1 inhibition. We also noted that vesicles in the W56 group were positioned further from the active zone (*Figure 5G*), indicating that Rac1 may modulate vesicle trafficking or dynamics. Despite the fact there was no difference in spine density, which is known to be regulated by postsynaptic processes involving Rac1 (*Costa et al., 2020*; *Runge et al., 2020*), a slight increase in spine area was observed in the W56 group (*Figure 5H*), possibly reflecting transsynaptic effects of altered presynaptic functions.

## Functional protein interaction network of presynaptic Rac1

To investigate the proteomic landscape of activity-dependent Rac1 interactions at presynaptic terminals, we employed in vivo proximity-dependent biotin identification (iBioID), a chemico-genetic proteomics technique that allows for spatially targeted protein identification using a promiscuous biotin ligase at synaptic sites (*Uezu et al., 2016*). We developed a presynaptic Rac1 BioID construct comprising a presynaptic protein synaptotagmin1, a small biotin ligase ultraID (*Kubitz et al., 2022*), and Rac1 in either constitutively active (CA) or dominant negative (DN) forms (*Figure 6A*). We validated the presynaptic localization of these fusion proteins in primary neuron cultures (*Figure 6B, C*) and expressed these constructs through AAV transduction in neonatal C57BL/6J mice (*n* = 9 per group). Following biotin injection, we harvested brain tissues at P28, isolated synaptosomes, and performed streptavidin bead pulldown to purify biotinylated proteins, which were analyzed via liquid chromatography–tandem mass spectrometry (LC–MS/MS) (*Figure 6D*). We validated the expression of the presynaptic Rac1 BioID constructs and biotinylation from the purified protein samples (*Figure 6E, F*).

In total, we quantified over 44,000 peptides corresponding to 3325 proteins. Using the advanced sensitivity of the newly released Orbitrap newly released Asymmetric Track Lossless (Astral) mass spectrometry analyzer, we were able to directly detect biotinylated peptides for 149 proteins in synaptosomes (*Figure 6—figure supplement 1A*), demonstrating their in vivo proximity to the BioID constructs at presynaptic terminals. To evaluate if these biotinylated proteins were indeed synaptic, we compared them with the synaptic gene datasets (*Koopmans et al., 2019*; *van Oostrum et al., 2023*). A high proportion (123/149) of biotinylated proteins were reported as enriched at synapses. Among the 97 SynGO-annotated proteins, 61 proteins were specifically localized at presynaptic terminals, including presynaptic membranes, vesicles, and active zones (*Figure 6—figure supplement 1B*), validating the spatial targeting of biotinylation. Additionally, a variety of proteins known to regulate or interact with Rac1, such as GEFs, GTPase-activating proteins (GAPs), and P21-activated kinases (PAKs), were present in the identified proteomes.

To assess which proteins preferentially interacted with active Rac1 (Rac1 CA), we filtered the biotinylated proteins based on the false discovery rate (FDR)-adjusted p-value and fold-change (FC) in enrichment. We identified 19 proteins that were enriched with the CA construct (*Figure 6G, H*). Gene ontology (GO) analysis revealed several functional clusters, including GTPases regulation (Arhgaps and Arhgefs), actin filament organization (Wasf3 and Abi2), and serine/threonine kinase activity (PAK1, PAK2, PAK3, and Mink1) (*Figure 6—figure supplement 1D*). The interaction between group I PAKs and active Rac1 aligns with previous studies that show binding of PAKs to GTP-bound Rac1 (*Byrne et al., 2016*; *Frost et al., 1996*; *Manser et al., 1994*), while the proximity of Mink1 to CA Rac1 presents a novel discovery because no previous studies have documented functional links between Mink1 and Rac1.

To explore whether interactions between these kinases and active Rac1 induces regulation via phosphorylation, we also used the advanced capability of the OrbiTrap Astral to identify phosphorylated peptides in the purified protein samples without TiO2 enrichment. We identified 495 phosphorylated peptides in total, among which 42 peptides corresponded to 22 proteins that were significantly enriched in the CA proteome. We further filtered these phosphopeptides to high-confidence

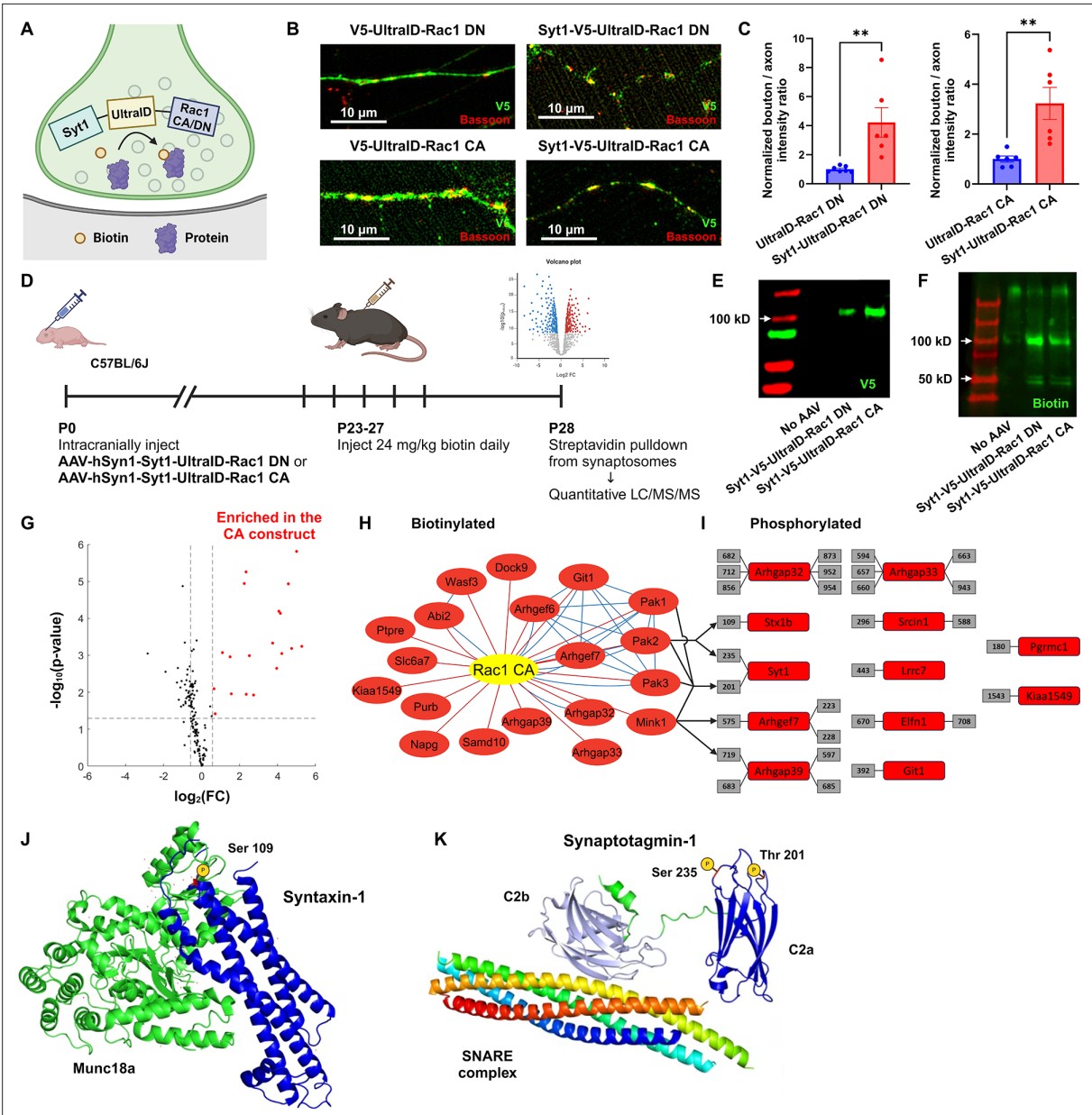

**Figure 6.** Functional protein interaction network of presynaptic Rac1. (**A**) Schematic of the presynaptic Rac1 BioID construct. (**B, C**) Presynaptic Rac1 BioID constructs are co-localized with a presynaptic marker Bassoon at synaptic boutons in C57BL/6J primary neuron cultures (UltraID-Rac1 DN: $n$ = 7 cells; Syt1-UltraID-Rac1 DN, UltraID-Rac1 CA, Syt1-UltraID-Rac1 CA: $n$ = 6 cells). (**D**) Timeline of AAV injection and protein sample collection. Expression of presynaptic Rac1 BioID constructs (**E**) and biotinylation (**F**) were verified using Western blot. (**G**) Volcano plot of the biotinylated proteins. Red dots represent the 19 proteins that were significantly enriched in the Rac1 constitutively active (CA) proteome. Dashed lines correspond to p-value = 0.05 (horizontal) or fold-change (FC) = ±1.5 (vertical). (**H**) Network of biotinylated proteins that were significantly enriched (FC >1.5 and p < 0.05) in the CA proteome. Edges represent protein–protein interactions identified from our proteomic data (red) or reported from STRING (blue). (**I**) Phosphorylated peptides that were significantly enriched in the CA proteome. Numbers in gray boxes represent phosphorylation sites. Arrows represent predicted kinase–substrate pairs with high percentile scores (>90). (**J**) Schematic of Syntaxin-1 and Munc18a (PDB ID: 3C98). A red residue represents the phosphorylation site (Ser 109). (**K**) Schematic of Synaptotagmin-1 and SNARE complex (PDB ID: 5CCG). Colors are assigned randomly for different proteins/domains. Data are expressed as mean ± standard error of the mean (SEM) with **p < 0.01.

The online version of this article includes the following figure supplement(s) for figure 6:

**Figure supplement 1.** Synaptic enrichment and additional analysis of the presynaptic Rac1 constitutively active (CA) and dominant negative (DN) proteomes.

candidates corresponding to proteins with identified biotinylated peptides, resulting in 29 peptides corresponding to 12 proteins (*Figure 6I*). We used a recently released kinase-substrate prediction tool, the Kinase Library v0.0.10 (*Johnson et al., 2023*), to test whether these proteins were phosphorylated by group I PAKs and/or Mink1 (*Figure 6—figure supplement 1E*). The percentile score of a kinase–substrate pair was above 90 for all kinases and one phosphorylated site (Syt1 Thr-201), for PAK2 and two sites (Stx1b Ser-109 and Syt1 Ser-235) and for Mink1 and two sites (Arhgap39 Ser-719 and Arhgef7 Thr-575). Mapping the phosphorylation sites of Stx1b and Syt1 to their structures (*Figure 6J, K*) reveals these sites sit within the three helical ($H_{abc}$) domain of Stx1b (S109) and the calcium-binding C2a domain of Syt1 (S235). These sites may be phosphoregulatory mechanisms to modulate synaptic vesicle dynamics. Hence, their enrichment in the CA proteome suggests that Rac1's presynaptic functions might be through phosphorylation. Importantly, both phosphorylation sites have previously been reported from synaptosomal phosphoproteomics (*Johnson et al., 2023*; *Trinidad et al., 2012*), validating they are *bona fide* phosphoacceptor sites. Overall, these findings suggest that presynaptic Rac1 interacts with kinases in an activity-dependent manner, which in turn regulates vesicle dynamics and GTPase activity through phosphorylation of proteins such as Stx1b and Syt1.

## Discussion

In this study, we have explored the specific role and regulatory mechanisms of the small Rho-family GTPase Rac1 at presynaptic terminals. Prior work using Rac1 knockout mice showed that Rac1 is important for working memory and longer-term forms of learning and memory. Our behavioral analyses reveal that these functions of Rac1 are likely segregated based on specific pre- and postsynaptic functions. We find that presynaptic Rac1 inhibition selectively impairs spatial working memory without affecting other memory forms. In contrast, postsynaptic Rac1 inhibition impairs remote memory. This is the first example we are aware of suggesting a synaptic compartmentalized role of Rac1 in hippocampus-dependent cognitive processes. Our EM analysis reveals that presynaptic Rac1 inhibition affects synaptic structures and the organization and morphology of synaptic vesicles, providing insights into presynaptic Rac1's role in vesicle pools. Mechanistically, our proteomic studies show the potential activity-dependent interaction of presynaptic Rac1 with key synaptic proteins and kinases, implicating the phosphorylation-mediated regulatory mechanisms involving synaptic vesicle dynamics and actin cytoskeleton remodeling that are consistent with the structural effects we observe by EM.

Our behavioral experiments consistently demonstrate that spatial working memory in mice is impaired by presynaptic Rac1 inhibition (*Figure 2*), suggesting its involvement in the cognitive processes regulating spatial navigation and memory retention over very short periods of time. Conversely, postsynaptic Rac1 inhibition does not affect working memory while exhibiting a subtle but nonsignificant effect on short term fear memory and a significant effect on remote memory (*Figure 3*). These data are generally consistent with prior Rac1 KO studies that also showed similar behavioral effects in working memory but were not able to distinguish the subcellular mechanisms of Rac1 function (*Haditsch et al., 2009*). Together, the data presented here highlight a site-specific function of Rac1 in synaptic modulation and its downstream effects on learning and memory, particularly spatial working memory. It is unlikely for motivational aspects to affect this process because both the exploratory behaviors in the radial-arm maze and Y-maze and the goal-directed activities in the T-maze with food restriction display impaired working memory performance.

The selective effect of presynaptic Rac1 on working memory, but not other types of memory, is intriguing given the hippocampus's association with various memory types, including short-term memory (STM) and LTM (*Izquierdo et al., 1999*). Our finding also challenges the conventional view that an intact working memory is a prerequisite for LTM formation. How can mice store and retrieve LTM while working memory is impaired? Prior research indicates that interlinked networks and brain regions control different memory types, each with partially shared or distinct regulatory mechanisms, such as neurotransmitter receptors (*Izquierdo et al., 1999*; *Sanderson and Bannerman, 2011*; *Shin et al., 2024*; *Yoon et al., 2008*). Although the extent of connectivity between distinct memory types is still unclear, accumulating data suggest the dissociation between spatial working memory and LTM. For example, rats lacking GluA1 subunit of AMPARs exhibit intact spatial reference memory but impaired spatial working memory in the radial-arm maze, Y-maze, and water maze tasks (*Bannerman et al., 2014*). Since STM and LTM are not affected by presynaptic Rac1 inhibition, our findings also

argue for the parallel processing of different memory types and that presynaptic Rac1 in the hippocampus is involved in allocentric-based spatial working memory but not in STM or LTM, specifically reference memory and fear memory.

Previous studies have reported conflicting results on the effects of altered Rac1 activity on learning and memory. Ablation of Rac1 in mature mouse hippocampal neurons is reported to impair working memory but have only subtle effects on LTM as evidenced by longer escape latency in a subset of acquisition sessions and no significant difference in the probe sessions in the MWM test (*Haditsch et al., 2009*). On the other hand, enhanced basal Rac1 activity in either BCR- or ABR-deficient mice was correlated to longer escape latency in a subset of acquisition sessions with only ABR$^{-/-}$ mice exhibiting impaired normal memory in probe sessions (*Oh et al., 2010*). Overexpression of Rac1 CA or Rac1 DN in the mouse dorsal hippocampus leads to faster or slower memory decay, respectively, in the novel object recognition task but no change in the contextual fear conditioning and trace fear conditioning (*Liu et al., 2016*). The discrepancies in these findings could be attributed to the diversity in methods for modulating Rac1 activity and the specific cellular or regional targets involved. In this study, to spatially target presynaptic compartments, we have inhibited Rac1 using the polypeptide W56, which has been utilized as a fusion protein by several studies to inhibit Rac1 in neurons spatially (*Hedrick et al., 2016*; *O'Neil et al., 2021*).

It is possible the impact on working memory might stem from alterations in short-term synaptic plasticity (STSP) induced by presynaptic Rac1 inhibition. Our previous findings indicated that in hippocampal neurons, presynaptic Rac1 is activated for approximately 2 min following an action potential (*O'Neil et al., 2021*), overlapping with the temporal dynamics of STSP and working memory processes. However, this was based on an overexpressed Rac1 FLIM sensor and thus the action potential (AP)-coupled activity time of endogenous Rac1 may be considerably shorter. Additionally, we demonstrated that presynaptic expression of W56 specifically influences the vesicle replenishment rate without affecting other synaptic characteristics, suggesting Rac1 might be selectively associated with STSP. While the link between STSP and working memory is still under investigation, computational models (*Fiebig and Lansner, 2017*; *Seeholzer et al., 2019*) and empirical studies (*Fujisawa et al., 2008*) have suggested their connection. Further research into presynaptic Rac1 and STSP in vivo could provide key insights into the mechanisms by which various types of synaptic plasticity influence cognitive functions and contribute to dissociation of different memory systems.

Our EM analysis highlights the impact of presynaptic Rac1 inhibition on postsynaptic structures and the morphology and distribution of synaptic vesicles (*Figure 5*). Interestingly, we have observed an enlargement in spine size following presynaptic Rac1 inhibition while spine density remains consistent. Previous research has documented Rac1's influence on both spine density and size with varying outcomes. The expression of CA Rac1 (RacV12) increases spine density and reduces spine size (*Tashiro et al., 2000*), whereas the expression of DN Rac1 (Rac1-N17) reduces both spine density and size in neurons (*Penzes et al., 2003*). Interestingly, spine size is increased 1 day after transfection of the Rac1-CA construct but decreased 6 days after transfection (*Hayashi-Takagi et al., 2010*). In our study, no significant difference in spine density is observed following presynaptic Rac1 inhibition, so spine density might be determined primarily by postsynaptic Rac1. In contrast to the aforementioned studies suggesting that both inactivation and prolonged activation of Rac1 result in smaller spines, we find that presynaptic Rac1 inhibition leads to an increase in spine size. It is possible that the transsynaptic effects induced by alterations in presynaptic Rac1 activity might have contributed to this change, as we have previously shown that presynaptic Rac1 inhibition reduces short-term synaptic depression (*O'Neil et al., 2021*). Additionally, the influence of presynaptic Rac1 on working memory appears to be independent of its known roles in synapse formation and maintenance (*Govek et al., 2005*), given that spine density remained constant.

One limitation of our analysis is the overexpression of Synapsin1a in both experimental and control groups, which might have obscured the specific effects of Rac1 inhibition on synaptic architecture. Although the impact of Synapsin1a overexpression on synaptic structures within the hippocampus is not well documented, evidence from other neural contexts indicates that it can significantly reduce vesicle numbers and size (*Vasileva et al., 2013*), which could explain why both the number of vesicles per bouton and vesicle size are relatively low even in the control group in the present study. However, it is unlikely to explain the differences between the scramble and W56 groups we observe, as both were fused to Synapsin1a.

Our functional proteomics mapping of presynaptic Rac1 has identified a select group of proteins known to interact with Rac1, confirming their interaction at presynaptic terminals as well (*Figure 6*). Among these, biotinylated proteins enriched in the CA proteome are characterized by several functional roles, including GTPase regulation, serine/threonine kinase activity, and actin cytoskeleton remodeling. The size of the proteomic dataset generated in this study is smaller than those typically reported in conventional proteomics research (*Uezu et al., 2016*; *van Oostrum et al., 2023*). This discrepancy arises from our specialized approach, which focuses exclusively on biotinylated proteins identified through novel Orbitrap-Astral high-resolution accurate mass spectrometry, offering a more precise view of proteins proximal to the BioID-tagged Rac1, thus ensuring greater spatial specificity in our findings.

We show that the functional role of presynaptic Rac1 and its signaling pathways may involve phosphoregulation via kinases coupled to actin cytoskeleton reorganization to regulate synaptic vesicle replenishment. One kinase we found is the serine/threonine kinase Misshapen-like kinase 1 (Mink1), which was significantly enriched in the CA proteome (FC = 2.15, adjusted p < 0.001). To our knowledge, no studies have yet reported the direct interaction between Rac1 and Mink1. A more likely explanation is that Mink1 and Rac1 are in proximity to each other at presynaptic terminals once Rac1 is activated. We also find that some of the phosphorylated and biotinylated proteins (Arhgap39, Arhgef7, and Syt1) are predicted to be phosphorylated by Mink1. Furthermore, MINK1 may phosphorylate Rho-GTPase regulators like ARHGAP21 and ARHGAP23 (*Daulat et al., 2022*), which affect Rac1 activity (*Zhang et al., 2016*). These data suggest Mink1's potential role in GTPase regulation and indirect involvement in a regulatory mechanism of presynaptic Rac1.

We also found the group I PAKs (PAK1/2/3), which are known downstream effectors of GTP-bound Rac1 (*Bokoch, 2003*; *Jacobs et al., 2007*; *Mierke et al., 2020*; *Rex et al., 2009*). The present study further supports that the Rac1/PAK signaling cascade is present at presynaptic terminals. Indeed, we found a phosphorylation site (S109) for syntaxin1b (Stx1b) that is within a short intrinsically disordered region linking two alpha helixes of syntaxin (*Huttlin et al., 2010*; *Trinidad et al., 2012*). This site is predicted to be a phosphorylation site for the PAK kinases based on a recently published atlas of kinase specificities (*Johnson et al., 2023*). Recently, it has become more evident that regulatory sites of phosphorylation map to short intrinsically disordered regions with folded domains (*Bludau et al., 2022*). Indeed, S109 is within a short loop linking to an alpha-helix of the $H_{abc}$ domain that interacts with Syntaxin-binding protein 1 (Munc18) (*Stepien et al., 2022*; *Figure 6J*). Thus, it is possible this

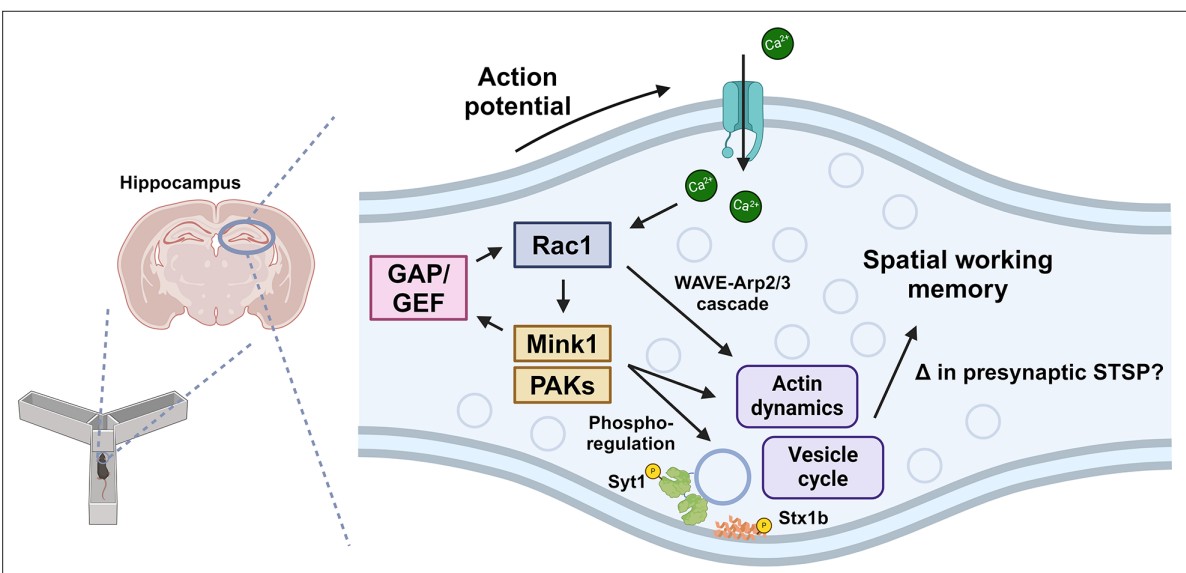

**Figure 7.** Model of presynaptic Rac1 regulation and its signaling cascades in spatial working memory. This study suggests this process is driven by kinase-mediated phosphorylation of Guanine nucleotide Exchange Factor (GEF)/GAPs and vesicle proteins, which regulate vesicle cycle and actin dynamics.

site, which is enriched in proximity to active Rac and PAK kinases in the presynaptic terminal, may influence synaptic vesicle dynamics through modulation of the syntaxin complex.

More intriguingly, we found two phosphorylation sites (T201, S235) in synaptotagmin-1 (syt1) that were also upregulated by active Rac1. While it is possible these are a consequence of our syt1 fusion probes for Rac activity, both sites have also been previously reported for endogenous syt1 from mouse synaptosomes (*Trinidad et al., 2012*) and are thus *bona fide* phosphoacceptor sites. In our previous work, we showed that Rac1 activity bidirectionally regulated presynaptic plasticity, with active Rac1 enhancing and inactive Rac1 reducing synaptic vesicle recycling rates. Synaptotagmins serve as a calcium sensor for membrane fusion of the vesicle (*Brose et al., 1992*; *Wu et al., 2021*), can regulate replenishment rates (*Liu et al., 2014*), and short-term presynaptic plasticity (*Bouazza-Arostegui et al., 2022*; *Chiu and Carter, 2024*; *Regehr, 2012*). Synaptotagmin-1 has two calcium-binding sites, C2a and C2b, and S235 sits within the calcium-binding pocket of C2a (*Figure 6K*). Mutational analysis of C2a suggests it may not be essential for synaptotagmin function, but instead has more subtle effects that alter hippocampal short-term plasticity (*Fernández-Chacón et al., 2002*). Indeed, we previously noted the roles of the related synaptotagmin-7 in regulating the complex interplay between calcium levels and vesicle availability that are linked to transient modulations of presynaptic plasticity (*Jackman et al., 2016*). Thus, our findings suggest the interesting possibility that phosphoregulation of synaptotagmin-1 C2a downstream of activated Rac1 may be a mechanism that explains our prior observations of its role in STSP. Whether this is linked to the current effects of Rac1 on working memory will require future work.

In summary, this study highlights the critical role of presynaptic Rac1 inhibition in selective modulation of spatial working memory, possibly mediated by phosphorylation of regulatory sites in the machinery that modulates vesicle availability and recycling (*Figure 7*). Future work on this mode of Rac1 signaling may provide insights into the dissociation of memory types and unravel the complex synaptic signaling pathways critical for cognitive processes and memory formation.

## Materials and methods

**Key resources table**

| Reagent type (species) or resource | Designation | Source or reference | Identifiers | Additional information |
|---|---|---|---|---|
| Strain, strain background (*Mus musculus*) | C57BL/6J | The Jackson Laboratory | Cat# 000664; RRID:IMSR_JAX:000664 | |
| Cell line (*Homo sapiens*) | HEK293T | ATCC | Cat# CRL-3216; RRID:CVCL_0063 | |
| Biological sample (*Mus musculus*) | Primary neuron cultures | This paper | | Freshly isolated from *Mus musculus* |
| Antibody | anti-GFP (Mouse monoclonal) | Synaptic Systems | Cat# 132 011; RRID:AB_887726 | ICC (1:500) |
| Antibody | anti-V5 (Mouse monoclonal) | Thermo Fisher Scientific | Cat# R960-25; RRID:AB_2556564 | IHC (1:500), WB (1:1000) |
| Antibody | anti-Bassoon (Rabbit polyclonal) | Synaptic Systems | Cat# 141 003; RRID:AB_887697 | ICC (1:500) |
| Antibody | anti-RFP (Mouse monoclonal) | Synaptic Systems | Cat# 409 011; RRID:AB_2800533 | ICC (1:500) |
| Antibody | anti-Homer1 (Rabbit polyclonal) | Synaptic Systems | Cat# 160 003; RRID:AB_887730 | ICC (1:500) |
| Antibody | anti-Gephyrin (Rabbit monoclonal) | Synaptic Systems | Cat# 147 008; RRID:AB_2619834 | ICC (1:500) |
| Antibody | anti-Biotin (Goat polyclonal) | Thermo Fisher Scientific | Cat# 31852; RRID:AB_228243 | WB (1:5000) |
| Recombinant DNA reagent | pAAV-hSyn1-W56-EGFP-Linker-Synapsin1a | *O'Neil et al., 2021* | | See Plasmids construction |
| Recombinant DNA reagent | pAAV-hSyn1-Scr-EGFP-Linker-Synapsin1a | *O'Neil et al., 2021* | | See Plasmids construction |

*Continued on next page*

*Continued*

| Reagent type (species) or resource | Designation | Source or reference | Identifiers | Additional information |
|---|---|---|---|---|
| Recombinant DNA reagent | pAAV-hSyn1-EGFP-Rac1 | *O'Neil et al., 2021* | | See Plasmids construction |
| Recombinant DNA reagent | pAAV-hSyn1-Syanptotagmin1-Linker-V5-UltraID-Linker-Rac1 CA | This paper | | See Plasmids construction |
| Recombinant DNA reagent | pAAV-hSyn1-Syanptotagmin1-Linker-V5-UltraID-Linker-Rac1 DN | This paper | | See Plasmids construction |
| Recombinant DNA reagent | pAAV-hSyn1-V5-UltraID-Linker-Rac1 CA | This paper | | See Plasmids construction |
| Recombinant DNA reagent | pAAV-hSyn1-V5-UltraID-Linker-Rac1 DN | This paper | | See Plasmids construction |
| Recombinant DNA reagent | pAAV-EF1a-W56-Linker-PSD95.FingR-eGFP-CCR5TC | This paper | | See Plasmids construction |
| Recombinant DNA reagent | pAAV-EF1a-Scr-Linker-PSD95.FingR-eGFP-CCR5TC | This paper | | See Plasmids construction |
| Recombinant DNA reagent | pAAV_Ef1a-W56-Linker-mScarlet-Gephyrin.FingR-IL2RGTC | This paper | | See Plasmids construction |
| Recombinant DNA reagent | pAAV_Ef1a-Scr-Linker-mScarlet-Gephyrin.FingR-IL2RGTC | This paper | | See Plasmids construction |
| Chemical compound, drug | Biotin | Millipore Signma | Cat# B4501; CAS: 58-85-5 | |
| Software, algorithm | Cytoscape | The Cytoscape Consortium | RRID:SCR_003032 | Version 3.9 |

## Mouse lines

All experiments were conducted in compliance with the National Institutes of Health guidelines and were approved by the Duke Institutional Animal Care and Use Committee (Protocol #: A144-23-07). C57BL/6J (Strain #:000664) mice were purchased from the Jackson Laboratory. Mice were housed in a group of 2–5 per cage and had free access to food and water under a 12-hr light/dark cycle.

## Primary neuron cultures

Prior to obtaining neuron cultures, 12 mm coverglasses (Neuvitro) were incubated overnight in 1 mg/ml poly-L-lysine (Sigma) solution in 24-well plates. Primary neuron cultures were prepared from P0 to P2 pups. The cortices and hippocampi were isolated in Hibernate-A medium (Gibco) at 4°C and then incubated in 5 mg/ml papain solution at 37°C for 18 min. Neurons were dissociated using a Pasteur pipet and plated on the coverglasses in 24-well plates at the concentration of 200,000 cells/well. Neurons were incubated in Neurobasal A medium (Thermo Fisher) supplemented with 10 µg/ml gentamicin, 2% B-27 (Gibco), 1% GlutaMAX (Gibco) at 37°C and 5% $CO_2$. At 4 days in vitro (DIV4), AAVs were added in neuron cultures at a multiplicity of infection of 150,000 viral genomes/cell, and 5 µM cytosine arabinoside (Sigma) was also added to suppress glial proliferation. Half of the medium was replaced with fresh medium once every 6–7 days.

## Plasmids construction

All plasmid constructs were verified prior to use by sequencing from either Eton Bioscience or Plasmidsaurus. pAAV-hSyn1-W56-EGFP-Linker-Synapsin1a, pAAV-hSyn1-Scr-EGFP-Linker-Synapsin1a, pAAV-hSyn1-DIO-TurboID, and pAAV-hSyn1-EGFP-Rac1 were previously generated in the Soderling lab.

pAAV-hSyn1-DIO-W56-EGFP-Linker-Synapsin1a was generated by replacing the TurboID region of pAAV-hSyn1-DIO-TurboID (EcoRI/NcoI sites) with the PCR product of the W56-EGFP-Linker-Synapsin1a region from pAAV-hSyn1-W56-EGFP-Linker-Synapsin1a (FWD and REV primers: 5'-GGCGCGCCCTAGAATTTCAGTCGGAGAAGAGGCTGGC-3' and 5'-ATGCTAGGCCACCATGATGGTGGACGGCAAGCCC-3', respectively) using In-Fusion cloning (TaKaRa Bio). pAAV-hSyn1-DIO-Scr-EGFP-Linker-Synapsin1a was generated by replacing the TurboID region of pAAV-hSyn1-DIO-TurboID (EcoRI/NcoI sites) with the PCR product of the Scr-EGFP-Linker-Synapsin1a region from

pAAV-hSyn1-Scr-EGFP-Linker-Synapsin1a (FWD and REV primers: 5′-GGCGCGCCCTAGAATTTCAG TCGGAGAAGAGGCTGGC-3′ and 5′-ATGCTAGGCCACCATGATGCTGCCCGGCTGGA-3′, respectively) using In-Fusion cloning.

pAAV-hSyn1-Syanptotagmin1-Linker-V5-UltraID-Linker-Rac1 CA and pAAV-hSyn1-Syanptotagmin1-Linker-V5-UltraID-Linker-Rac1 DN were generated by replacing the EGFP-Rac1 region of pAAV-hSyn1-EGFP-Rac1 (BamHI/NcoI sites) with the PCR products of the gene fragments (Twist Bioscience) encoding Syanptotagmin1-Linker-V5-UltraID-Linker-Rac1 CA or Syanptotagmin1-Linker-V5-UltraID-Linker-Rac1 DN, respectively, using multiple-insert In-Fusion cloning. The FWD and REV primers for gene fragment 1 were 5′-TAGAGTCGACACCATGGTGAGTGCCAGTCGT-3′ and 5′-TTAAAGTCGCTA GCGGTGCTGTCCAGG-3′, respectively, and the FWD and REV primers for gene fragment 2 were 5′-CCGCTAGCGACTTTAAGAATCTAATTTGGTTGAAAGAAGTGGAT-3′ and 5′-ATAAGCGAATTGGATC CTTACAACAGCAGGCATTTTCTCTTCCTCT-3′. Rac1 CA was encoded by IKCVVVGDGAVGKTCLLISY TTNAFPGEYIPTVFDNYSANVMVDGKPVNLGLWDTAGLEDYDRLRPLSYPQTDVFLICFSLVSPASFHHVRA KWYPEVRHHCPNTPIILVGTKLDLRDDKDTIEKLKEKKLTPITYPQGLAMAKEIGAVKYLECSALTQRGLKT VFDEAIRAVLCPPPVKKRKRKCLLL. Rac1 DN was encoded by IKCVVVGDGAVGKNCLLISYTTNAFPGE YIPTVFDNYSANVMVDGKPVNLGLWDTAGQEDYDRLRPLSYPQTDVFLICFSLVSPASFENVRAKWYP EVRHHCPNTPIILVGTKLDLRDDKDTIEKLKEKKLTPITYPQGLAMAKEIGAVKYLECSALTQRGLKTVFDE AIRAVLCPPPVKKRKRKCLLL.

pAAV-hSyn1-V5-UltraID-Linker-Rac1 CA and pAAV-hSyn1-V5-UltraID-Linker-Rac1 DN were generated by replacing the EGFP-Rac1 region of pAAV-hSyn1-EGFP-Rac1 (BamHI/NcoI sites) with the PCR products of the gene fragments (Twist Bioscience) encoding V5-UltraID-Linker-Rac1 CA (FWD and REV primers: 5′-TAGAGTCGACACCATGGGCAAGCCCATCCCCAACC-3′ and 5′-ATAAGCGAATTG GATCCTTACAACAGCAGGCATTTTCTCTTCCTCT-3′, respectively) or V5-UltraID-Linker-Rac1 DN (FWD and REV primers: 5′-CCGCTAGCGACTTTAAGAATCTAATTTGGTTGAAAGAAGTGGAT-3′ and 5′-ATAAGCGAATTGGATCCTTACAACAGCAGGCATTTTCTCTTCCTCT-3′, respectively) using In-Fusion cloning.

pAAV_hSyn1_DIO_Syanptotagmin1-Linker-V5-UltraID-Linker-Rac1 CA and pAAV_hSyn1_DIO_Syanptotagmin1-Linker-V5-UltraID-Linker-Rac1 DN were generated by replacing the TurboID region of pAAV-hSyn1-DIO-TurboID (EcoRI/NcoI sites) with the PCR products of the Syanptotagmin1-Linker-V5-UltraID-Linker-Rac1 CA region from pAAV-hSyn1-Syanptotagmin1-Linker-V5-UltraID-Linker-Rac1 CA or the PCR products the Syanptotagmin1-Linker-V5-UltraID-Linker-Rac1 DN region from pAAV-hSyn1-Syanptotagmin1-Linker-V5-UltraID-Linker-Rac1 DN, respectively, using In-Fusion cloning. The FWD and REV primers were 5′-ATGCTAGGCCACCATGGTGAGTGCCAGTCG-3′ and 5′-GGCGCGCC CTAGAATTCCAACAGCAGGCATTTTCTCTTCCTCTTCTTCA-3′, respectively.

pAAV-EF1a-W56-Linker-PSD95.FingR-eGFP-CCR5TC and pAAV-EF1a-Scr-Linker-PSD95.FingR-eGFP-CCR5TC were generated by replacing the BamHI/CsiI sites of pAAV-EF1a-PSD95.FingR-eGFP-CCR5TC (Addgene #125691) with the PCR products of the gene fragments (Twist Bioscience) encoding W56-PSD95.FingR or Scr-PSD95.FingR, respectively, using In-Fusion cloning. The FWD and REV primers were 5′-GTACCGAGCTCGGATCCGCCGCCACCATG-3′ and 5′-TATAGTCGACACCAGG TTTCAGGCCGCTG-3′, respectively.

pAAV_Ef1a-W56-Linker-mScarlet-Gephyrin.FingR-IL2RGTC and pAAV_Ef1a-Scr-Linker-mScarlet-Gephyrin.FingR-IL2RGTC were generated by replacing the NcoI/BstEII sites of pAAV-EF1A-mScarlet-Gephyrin.FingR-IL2RGTC (Addgene #125695) with the PCR products of the gene fragments (Twist Bioscience) encoding W56-mScarlet (FWD and REV primers: 5′-TAAGCTTGCCACCATGGTGGACGG CAAGC-3′ and 5′-GGCCACCCTTGGTCACCTTCAGCTTGGCGG-3′, respectively) or Scr-mScarlet (FWD and REV primers: 5′-TAAGCTTGCCACCATGCTGCCCGGCTGG-3′ and 5′-GGCCACCCTTGG TCACCTTCAGCTTGGCGG-3′, respectively) using In-Fusion cloning.

## Virus production

Large-scale AAVs were produced as previously described (*Uezu et al., 2016*). Briefly, HEK293T (human embryonic kidney 293T) cells were incubated in Dulbecco's modified Eagle medium (DMEM) medium (Gibco) supplemented with 10% fetal bovine serum (Sigma-Aldrich) and 1% penicillin–streptomycin (Gibco) at 37°C and 5% $CO_2$. For each virus, HEK293T cells were plated on six 15 cm plates at a density of $1.5 \times 10^7$ cells/plate. After 24 hr, cells were transfected with 15 µg AAV plasmid with the transgene, 15 µg serotype plasmid AAV2/9, and 30 µg pAdDF6 using PEI MAX (Polysciences). At 72 hr

after transfection, the cells were collected, resuspended in 4 ml of cell lysis buffer (15 mM NaCl, 5 mM Tris–HCl, pH 8.5), and freeze-thawed three times. The cell lysate was treated with 50 U/ml benzonase (Millipore) and centrifuged at 3300 × $g$ for 30 min. The supernatant was added on the gradient of 15%, 25%, 40%, and 60% iodixanol solution and ultracentrifuged at 370,000 × $g$ for 90 min using a Beckman Ti-70 rotor. The viral solution at the interface of the 40% and 60% iodixanol gradient was collected, washed with 1× phosphate-buffered saline (PBS) in a 100 kDa filter (Millipore) at least three times and until the final volume is less than 200 µl. Purified AAVs were aliquoted and stored at −80°C.

Small-scale AAVs were produced as previously described (*O'Neil et al., 2021*). Briefly, for each virus, HEK293T cells were plated on a 12-well plate at a density of $2.5 \times 10^5$ cells/well. After 24 hr, cells were transfected with 0.4 µg AAV plasmid with the transgene, 0.4 µg serotype plasmid AAV2/9, and 0.8 µg pAdDF6 using PEI MAX (Polysciences). At 6–18 hr after transfection, all medium was aspirated and replaced with 1 ml DMEM without glutamine (Gibco) supplemented with 10% fetal bovine serum and 1% GlutaMAX. At 72 hr after transfection, the medium was collected, centrifuged in a 0.45-µm Spin-X filter at 2350 × $g$ for 1 min, and stored at 4°C until use.

## Stereotaxic injection of AAV

Adult mice at P35–42 were stereotaxically injected with high-titer (>$10^{13}$ VG/ml) AAVs bilaterally in the hippocampus. Mice were anesthetized with isoflurane. Microinjection dispense system, Picospritzer III (Parker), was used to inject small volume (~220 nl per site) of AAVs in two sites per hemisphere (AP −1.46, ML ±1.25, DV −2; and AP −2.46, ML ±2.4, DV −2.3) at a rate of 50 nl/min. The microinjection needle was inserted into the brain slowly at a speed of 0.5 mm/min and was retracted 5 min after AAV injection. Body weights of mice were recorded before and 24 hr after the surgery.

## Mouse behavioral experiments

All behavioral experiments were performed in the Duke Mouse Behavioral and Neuroendocrine Core Facility for behavioral tests. Food and water were provided ad libitum unless specified. To minimize the effects of olfactory cues, all apparatus were thoroughly wiped with disinfectant (LabSan 256 CPQ, Sanitation Strategies) before and in-between tests. Behavioral experiments were conducted with a minimum interval of 1 week between each experiment. The experimenters were blinded to the treatment groups to ensure objective assessment.

### Light/dark transition test

The light/dark transition test was conducted in a rectangular, two-compartment box. One compartment was brightly illuminated by a white bulb, while the other compartment was dark and covered with an opaque lid. The compartments were connected by a sliding door controlled by Med-PC Behavioral Control Software. Each mouse was placed in the dark compartment and allowed to move freely between the two compartments for 5 min. The Behavioral Control Software was used to record the time spent in each compartment, the number of transitions between compartments, and the total distance traveled.

### Open field test

Mice were placed in a square open field chamber (Omnitech Electronics Inc) and allowed to explore for 5 min. Locomotor activity was recorded using a camera on top and photosensors on sides of each chamber and analyzed using the Fusion software (Omnitech Electronics Inc). A horizontal activity count is a count of sensor changes (beam breaks), and a vertical activity count is a count of vertical beam breaks. A stereotypic activity count is the number of beam breaks due to a stereotypic activity, which is counted when a mouse breaks the same beam or the same set of beams repeatedly. Episodes are defined as periods of activity separated by rest periods of at least 1 s.

### Radial-arm maze test

The radial-arm maze was made of eight identical arms conjoined at the center. Each arm was flanked by walls on both sides for the initial one-third of their length from the center of the maze. The maze was elevated 50 cm above the floor. The maze was surrounded by curtains with visual cues. Mice were placed at the center of the radial-arm maze and allowed to explore for 5 min. Video recordings were analyzed using a manual scoring method, where the specific arm and time point were recorded when

all four paws of a mouse crossed beyond the walls. An arm entry was considered as an 'error' if the arm was previously visited in the given 5-min session. The percentage of errors was calculated as the number of errors divided by the total number of arm entries.

## Y-maze test

The Y-maze was made of three identical closed arms conjoined at the center. The maze was surrounded by curtains with visual cues. Mice were placed in the center of the Y-maze and allowed to explore for 5 min. Activity was recorded using a camera above the maze and analyzed using a video tracking software (Noldus EthoVision XT). An alternation was defined as the entry into the three arms consecutively. The percentage of alternation was calculated as the number of alternations divided by the number of maximum alternations.

## DNMTP T-maze test

Prior to the DNMTP test, mice ($n$ = 8–10 per treatment group) were placed on a food restriction protocol to maintain them at 85–90% of their free-feeding body weight. Water was available ad libitum. Body weight was monitored daily to adjust the food ration and maintain the desired body weight.

The T-maze was made of two goal arms and a longer start arm conjoined in the shape of 'T'. The maze was elevated 50 cm above the floor. At the end of each goal arm, a food well was present where a food reward could be placed. The maze was surrounded by curtains with visual cues. The test consisted of habituation, pre-training, and the DNMTP task. During habituation, a small amount of food reward was mixed in rodent diets, and mice were handled for 1 min daily for 6 consecutive days. Pre-training involved six forced trials daily per mouse. In the forced trial, one of the goal arms was pseudo-randomly chosen (three times each for the left or right goal arm, in a random order) and blocked, and a food reward was placed at the end of the open goal arm. A mouse was placed in the start arm and allowed to explore the T-maze until they consumed the reward or 30-s period passed. The DNMTP task was conducted after all mice consumed food rewards in at least five forced trials during the pre-training. The DNMTP task involved six pairs of a forced trial and a choice trial with an ITI of 5 s daily for 7 days. If a mouse consumed the food reward in the forced trial, the mouse was removed from the maze, and a choice trial was performed after 5 s. In the choice trial, the mouse was placed back at the start arm with both goal arms open with a food reward at the arm opposite to the open goal arm in the forced trial. A correct choice was recorded if the mouse entered the correct arm on the first attempt. Upon entering an incorrect arm, the mouse was confined in the arm for 10 s. The percentage of correct choices was calculated as the number of correct choice trials divided by the total number of choice trials for every mouse each day. A percentage of correct choices was not included if a mouse failed five or all forced trials on a given day. The Tukey multiple comparison test was performed to compare means of correct choices percentages across treatment groups.

## MWM test

Prior to water maze testing, mice were handled for 1 min daily for 5 days. Subsequently, on days 6 and 7, mice were placed in a cage filled with shallow water of 1 cm depth for 30 s. On day 8, mice were allowed to freely swim in deep water for 30 s. On days 9 and 10, mice were placed on a platform submerged 1 cm below the water surface for 5 s and subsequently allowed to swim freely until they climbed onto the platform or 30 s elapsed.

The water maze apparatus consisted of an open circular pool, filled halfway with opaque water. Activity was recorded using a camera above the maze and analyzed using a video tracking software (Noldus EthoVision XT).

Two pairs of acquisition trials were performed each day for 6 consecutive days. The ITI within a pair was 5 s, and ITI between two pairs was 30 min. In the acquisition trial, a circular metal platform (15 cm in diameter) was submerged 1 cm below the water surface and placed in one of the four quadrants, Northeast (NE), Northwest (NW), Southeast (SE), or Southwest (SW). The platform remained in the same position for the duration of testing for all mice. For each acquisition trial, the starting position where the mouse is placed in the pool was chosen pseudo-randomly among eight positions (N, E, S, W, NE, NW, SE, and SW), excluding the position with the hidden platform. The acquisition trial ended when the mouse climbed onto the platform or when 60 s elapsed.

Probe trials were conducted 30 min after the last acquisition trial on days 2, 4, and 6. The hidden platform was temporarily removed from the pool prior to the probe trials. Each mouse was placed opposite from the original position of the platform and allowed to swim freely for 60 s. Percentage of time spent in the platform quadrant was calculated as time spent in the quadrant where the hidden platform had been located during the acquisition trials divided by 60 s, the total duration of a probe trial.

## Fear conditioning test

In a training session, mice were placed in a chamber with a metal grid floor. Mice were allowed to explore for 2 min, after which an auditory tone was played as a conditioned stimulus (CS) for 30 s. Immediately after the CS, an electric shock (0.4 mA, 2 s) was applied as an unconditioned stimulus (US). After the shock, mice were allowed to explore for 30 s.

Contextual fear conditioning sessions were conducted 1 hr, 24 hr, and 10 days after the training session. In a contextual fear conditioning session, mice were placed in the same chamber and allowed to explore for 5 min.

Cued fear conditioning session was conducted 48 hr after the training session. In a cued fear conditioning session, mice were placed in a chamber with an altered context, including dimmer room light, walls with different colors, and novel objects placed inside the chamber. Mice were allowed to explore for 5 min, where the same auditory tone in the conditioning session was played for the last 3 min.

For all sessions, mouse activity was recorded using a camera installed on the top of each chamber. Freezing time, defined as a motionless period lasting 1 s or more, was calculated using a fear conditioning software, CleverSys FreezeScan. Chambers were thoroughly cleaned with sanitizer after every session.

## Shock sensitivity threshold evaluation

The assessment of sensitivity to scrambled foot shock was conducted using a Med-Associates (St. Albans, VT) startle platform. Mice were placed in a Plexiglas enclosure featuring a grid floor, with each grid linked to an electric harness connected to a scrambler module (Med-Associates), which generated foot shocks of different intensities. The procedure consisted of a 2-min period for acclimatization, followed by 10 trials where mice received shocks (250 ms) with ITIs of 20–90 s. Shock intensities across the 10 trials included 0, 0.1, 0.2, 0.3, 0.4, 0.5, and 0.6 mA. The startle response of each mouse was recorded using piezoelectric sensors during the initial 1000 ms following the onset of the shock stimulus.

## Immunocytochemistry

Prior to immunocytochemistry, fixation buffer (4% PFA (Sigma-Aldrich), 4% sucrose in 1× PBS), permeabilization/wash buffer (0.1% Triton X-100 in 1× PBS) and blocking buffer (5% normal goat serum (Jackson ImmunoResearch), 0.1% Triton X-100 in 1× PBS) were prepared. Primary neuron cultures were fixed in the fixation buffer at 4°C for 10 min and were gently shaken in 1× PBS two times at room temperature (RT) for 5 min, in the permeabilization buffer at RT for 7 min, in the blocking buffer at RT for 1 hr, in the blocking buffer with primary antibodies at 4°C overnight, in the wash buffer three times at RT for 5 min, in the blocking buffer with secondary antibodies at RT for 1 hr, and in the wash buffer three times at RT for 5 min. The coverglass with the neuron cultures was transferred onto a glass slide with FluorSave Reagent (Millipore).

## Transcardiac perfusion and immunohistochemistry

For transcardiac perfusion, mice were anesthetized with isoflurane until there was no response to tail or toe pinches. In a chemical fume hood, mice were transcardially perfused with saline for 10 min and 4% paraformaldehyde (PFA) in 1× PBS for 30 min at a speed of 6.5 ml/min. The brain was harvested and stored in 4% PFA in 1× PBS at 4°C for 1–2 days until use.

Prior to immunohistochemistry, wash buffer (0.1% Triton X-100 in 1× PBS) and blocking buffer (5% normal goat serum, 0.2% Triton X-100 in 1× PBS) were prepared. Fixed brains were sectioned either coronally or sagittally at a thickness of 50 µm using a vibratome (Leica Biosystems). Brain slices were gently shaken in 1× PBS three times at RT for 5 min, in the blocking buffer at RT for 2 hr, in the blocking buffer with primary antibodies at 4°C overnight, in the wash buffer three times at RT for

10 min, in the blocking buffer with secondary antibodies at RT for 1 hr, and in the wash buffer three times at RT for 10 min. The brain slices were transferred onto a glass slide with FluorSave Reagent.

Antibodies used and the corresponding dilutions were as follows: mouse-anti-GFP (1:500, Synaptic Systems 132011), mouse-anti-V5 (1:500 for IHC and 1:1000 for WB, Thermo Fisher R960-25), rabbit-anti-Bassoon (1:500, Synaptic Systems 141003), mouse-anti-RFP (1:500, Synaptic Systems 409011), rabbit-anti-Homer1 (1:500, Synaptic Systems 160003), rabbit-anti-Gephyrin (1:500, Synaptic Systems 147008), goat-anti-biotin (1:5000, Invitrogen 31852), goat-anti-mouse 488 (1:1000, Invitrogen A32723), goat-anti-rabbit 637 (1:1000, Invitrogen A32733), goat-anti-mouse 800 (1:5000, LI-COR 926-32210), and rabbit-anti-goat horseradish peroxidase (HRP)-conjugate (1:5000, Bio-Rad 1721034).

## In vivo proximity-dependent biotin identification

High-titer AAVs (>$10^{13}$ VG/ml, 2 µl per hemisphere) were intracranially and bilaterally injected into P0-1 C57BL/6J pups under hypothermia anesthesia. Pups were placed under a heat lamp until recovery and then returned to their home cage. After 23–27 days post-AAV injection, mice were subcutaneously injected with 24 mg/kg biotin on a daily basis for 5 consecutive days. At P28, the cortices and hippocampi were harvested, flash-frozen using liquid nitrogen, and stored at −80°C.

Prior to synaptosome purification, homogenization buffer (320 mM sucrose, 5 mM 4-(2-hydroxyethyl)-1-piperazineethanesulfonic acid (HEPES), 1 mM ethylene glycol tetraacetic acid (EGTA), pH 7.4), resuspension buffer (320 mM sucrose, 5 mM Tris/HCl, pH 8.1), and sucrose solutions (0.8, 1, and 1.2 M sucrose with 5 mM HEPES) were prepared. On the day of purification, all solutions were supplemented with cOmplete protease inhibitor (Roche) and PhosSTOP phosphatase inhibitor (Roche) cocktails. The brain was thawed in homogenization buffer and homogenized using a Dounce tissue homogenizer. The homogenized tissues were centrifuged at 1000 × $g$ for 10 min. The supernatant was collected and centrifuged at 12,000 × $g$ for 20 min. The pellet was resuspended in resuspension buffer and transferred onto a discontinuous sucrose gradient (1.2/1/0.8 M sucrose solutions) in a Beckman Ultra-Clear 14 × 89 mm tube. This gradient was then ultracentrifuged at 85,000 × $g$ for 2 hr. The synaptosomal plasma membrane fraction at the interface of 1 and 1.2 M sucrose solutions were collected, transferred onto 0.8 M sucrose solution, and ultracentrifuged at 85,000 × $g$ for 1 hr. The pellet was stored −80°C until use.

Prior to the streptavidin pulldown, radioImmunoprecipitation assay (RIPA) buffer (50 mM Tris/Cl, 150 mM NaCl, 1 mM ethylenediaminetetraacetic acid [EDTA], 0.4% sodium dodecyl sulfate [SDS], 2% Triton X-100, 2% deoxycholate) and lysis buffer (50 mM Tris/Cl, 150 mM NaCl, 1 mM EDTA), and elution buffer (2% SDS, 25 mM Tris, 50 mM NaCl, 10 mM DTT, 2.5 mM biotin) were prepared. On the day of purification, all solutions were supplemented with cOmplete protease inhibitor and PhosSTOP phosphatase inhibitor cocktails. Synaptosomes were thawed and pooled (3 brains per replicate; 3 replicates per AAV construct) in an equal mix of lysis and RIPA buffers, followed by sonication for three 20-s intervals. After adding SDS to a final concentration of 1%, the lysates were heated in boiling water for 5 min. The lysates were then cooled on ice and combined with 50 µl of washed Pierce Protein A Agarose (Thermo Fisher). The mixture was rotated at 4°C for 30 min and centrifuged at 3,000 × $g$ for 1 min. The supernatant was carefully collected and then combined with 50 µl of Pierce NeutrAvidin agarose (Thermo Fisher), followed by rotation at 4°C overnight. After centrifugation at 3000 × $g$ for 1 min, the bead pellet was transferred to a low-protein-binding tube and washed twice with 2% SDS, twice with 25 mM LiCl/1% deoxycholate/1% Triton X-100, twice with 1 M NaCl, and five times with 50 mM Ambic in MS-grade water. The tube was placed on a nutator at room temperature for 10 min between washing steps. After washing, the biotinylated proteins were eluted from the beads by adding 50 µl of elution buffer and heating at 95°C for 5 min. After centrifugation at 3000 × $g$ for 1 min, the supernatant containing the biotinylated proteins was transferred to a new low-protein-binding tube and stored at −80°C.

## Electrophoresis and Western blotting

Equal amounts (10 µl) of the eluted solution from the streptavidin pulldown were loaded in a 4–20% polyacrylamide gel (Bio-Rad 4561096) and separated with SDS–polyacrylamide gel electrophoresis at 100 V for 1 hr. After electrophoresis, proteins were transferred to a nitrocellulose membrane (Cytiva 10600002) at 90 V for 75 min. After transfer, the membranes were blocked for 1 hr using Intercept Blocking buffer (LI-COR 927-60001) or 5% milk in 1× TBST (TBS with 0.05% Tween-20). The

membranes were then incubated with primary antibodies (mouse-anti-V5 in Intercept Blocking buffer or goat-anti-biotin in 5% milk 1× TBST) at 4°C overnight. The membranes were washed three times with TBST and incubated with secondary antibodies (goat-anti-mouse 800 in Intercept Blocking buffer or rabbit-anti-goat HRP-conjugate in 5% milk 1× TBST) at RT for 1 hr. After washing three times with TBST, the membranes were imaged using the Odyssey Imaging System and the Image Studio Ver 5.2.

## Differential protein expression analysis by quantitative LC–MS/MS

The protein samples were kept in dry ice and sent to the Duke Proteomics and Metabolomics Core Facility for LC–MS/MS. The samples were initially spiked with either 1 or 2 pmol of bovine casein, serving as an internal quality control. They were adjusted to a 5% SDS concentration, reduced at 80°C for 15 min, and alkylated with 20 mM iodoacetamide at room temperature for 30 min, followed by addition of 1.2% phosphoric acid and 375 µl of S-Trap (Protifi)-binding buffer (90% methanol, 100 mM triethylammonium bicarbonate [TEAB]). The proteins were trapped on the S-Trap micro cartridge, digested with 20 ng/µl sequencing grade trypsin (Promega) at 47°C for 1 hr, and eluted sequentially with 50 mM TEAB, then 0.2% formic acid, and finally a mixture of 50% acetonitrile with 0.2% formic acid. Post-elution, the samples were lyophilized until dry and then reconstituted in 1% trifluoroacetic acid (TFA)/2% acetonitrile containing 12.5 fmol/µl of yeast alcohol dehydrogenase (ADH). Equal volumes from each sample were combined to produce a study pool quality control.

Quantitative LC/MS/MS analysis was conducted on 2 µl of each sample using a Vanquish Neo UPLC system (Thermo) linked to an Orbitrap Astral high-resolution accurate mass tandem mass spectrometer (Thermo). The sample was trapped on a Symmetry C18 20 mm ×180 µm trapping column, with a flow rate of 5 µl/min in a 99.9/0.1 vol/vol water/acetonitrile solution. The analytical separation was executed on a 1.5-µm EvoSep 150 µm ID × 8 cm column using a 30-min gradient of 5–30% acetonitrile with 0.1% formic acid, at a flow rate of 500 nl/min and a column temperature of 50°C. The Orbitrap Astral mass spectrometer collected data in a data-independent acquisition (DIA) mode, with a resolution of 240,000 at mass-to-charge ratio ($m/z$) of 200 for full MS scans from $m/z$ 380 to 980, targeting 4e5 ions for an automatic gain control (AGC). Fixed DIA windows of 4 $m/z$ from $m/z$ 380 to 980 DIA MS/MS scans were acquired, targeting AGC value of 5e4 and a maximum fill time of 6 ms, using a 27% HCD collision energy for all MS2 scans. The total cycle time for each sample injection was approximately 35 min.

After UPLC–MS/MS analyses, the data were imported into Spectronaut (Biognosis) and aligned individually based on the accurate mass and retention time of detected precursor and fragment ions. Relative peptide abundance was quantified from MS2 fragment ions in the selected ion chromatograms of aligned features across all runs. The MS/MS data were searched against the SwissProt *M. musculus* database (downloaded in 2022), a common contaminant/spiked protein database, and an equivalent number of reversed-sequence 'decoys' for FDR evaluation. Database searches were performed using a library-free approach within Spectronaut. Search parameters included a fixed modification on Cys (carbamidomethyl) and a variable modification on Met (oxidation). Full trypsin digestion rules were applied with mass tolerances of 10 ppm for precursor ions and 20 ppm for product ions. Spectral annotation was confined to a maximum 1% peptide FDR, based on $q$-value calculations. Peptide homology was addressed using razor rules, assigning a peptide matched to multiple proteins exclusively to the one with the most identified peptides. Protein homology was assessed by grouping proteins sharing identical peptide sets, assigning a master protein within each group based on percentage of coverage.

The raw intensity values for each precursor were generated by the Spectonaut detection software. No value was assigned when the peak detection criteria were not met. The missing values were imputed as follows: the values for peptides with the same sequence but different precursor charge states were summed into a single peptide value. Subsequently, if less than half of the values are missing within a biological group, missing values were imputed with intensities derived from a normal distribution defined by measured values within the same intensity range (20 bins). If more than half of the values were missing and the peptide intensity exceeded 5e6, the intensity was set to 0. Any remaining missing values were imputed with the lowest 2% of all detected values. Before normalization, peptides not measured at least twice across all samples and in at least 50% of the replicates within any single group were removed. Total intensity normalization was performed such that total intensity of all peptides for a sample was summed and normalized across all samples. Subsequently,

trimmed-mean normalization was performed to exclude the top and bottom 10% of signals, and the average of the remaining values was used to normalize across all samples. Values for all peptides belonging to the same protein were combined into a single intensity value.

Proteins from different species and contaminants, such as keratins and avidins, were removed from the dataset. FC was defined as a ratio of protein intensity in the experimental sample to that in the control sample, and FDR-adjusted p-value was calculated from two-tailed heteroscedastic $t$-tests on log2-transformed protein intensities. Proteins were considered significantly enriched in the experimental sample if their FC was higher than 1.5 and FDR-adjusted p-value lower than 0.05. Protein–protein interaction (PPI) networks were generated using Cytoscape 3.10.1. Each node corresponds to a gene name, and an edge between nodes shows a PPI. The previously known interactions were identified using the STRING database (v12.0) with a minimum required interaction score of 0.7. The enriched GO terms were identified using ShinyGO v0.80, and synaptic localization was annotated using SynGO v1.1.

## Electron microscopy

C57BL/6J mice (4 months old) were stereotaxically injected with AAV-hSyn1-W56-Synapsin1a or AAV-hSyn1-Scr-Synapsin1a (3 mice each; 6 mice in total) bilaterally in the hippocampus. The procedure and coordinates were as described in the Stereotaxic injection of AAV section. After 1 month post-injection, mice were transcardially perfused with cold physiological saline for 2 min, until blood is washed out and then with fixative (0.1% EM-grade glutaraldehyde (GA), 4% PFA in 0.1 M phosphate buffer (PB, pH 7.2)) for 45 min at a flow rate of 6.5 ml/min. Brains were post-fixed in GA-free 4% PFA in 0.1 M PB at 4°C overnight. Sections with a thickness of 70 µm were prepared using a Leica vibratome. No noticeable variations in the size of the hippocampus were observed across experimental groups.

For EM, sections were post-fixed with 1% osmium tetroxide, subjected to a graded ethanol dehydration process, and then embedded in Durcupan epoxy resin (Sigma, Germany) within Aclar sheets (EMS, Hatfield, PA, USA). Rectangular samples, precisely cut from the dorsal region of the CA1 hippocampus (approximately −4.0 mm from bregma), were prepared under a Leica S6D dissecting microscope and mounted onto plastic blocks. Using a Reichert ultramicrotome, ultrathin sections of 60 nm were cut, placed onto 300 mesh copper grids, stained with lead citrate (Ultrostain II, Leica), and then analyzed using a JEM-1011 transmission electron microscope (JEOL, Tokyo, Japan). This microscope was equipped with a Mega-View-III digital camera and a Soft Imaging System (SIS, Münster, Germany). For each block, 5–10 sections were examined, with two blocks analyzed per animal to ensure a comprehensive collection of micrographs. Sample areas (>50 µm² for each animal) were selected semi-randomly. Electron micrographs were used to identify postsynaptic dendritic spines, axonal boutons, and mitochondrial profiles. The areas of spine and mitochondrion profiles were quantified using NIH ImageJ v1.49o (*Schneider et al., 2012*). Data were organized and analyzed with KaleidaGraph (Synergy Software, Reading, PA, USA) software. Blind data collection and analysis were conducted to prevent any bias.

## Subcellular localization quantification

Subcellular localization of protein constructs was quantified as previously described (*O'Neil et al., 2021*). Briefly, primary neuron cultures ($n$ = 3 coverglasses per group) were imaged using a lattice structured illumination microscopy (Zeiss Elyra 7) with a Plan-Apochromat ×63/1.4 oil objective lens. Maximum intensity projection was calculated from Z-stack images. Areas corresponding to presynaptic/postsynaptic sites, axons/dendrites, and background were manually outlined using ImageJ on channels corresponding to Bassoon, Homer1, or Gephyrin. Bouton/axon intensity ratio and postsynaptic/dendritic intensity ratio were calculated as $(Intensity_{bouton} − Intensity_{background})/(Intensity_{axon} − Intensity_{background})$ and $(Intensity_{postsynaptic} − Intensity_{background})/(Intensity_{dendritic} − Intensity_{background})$, respectively.

## Statistical analyses

No statistical methods were used to determine sample sizes. Statistical analysis was performed in Prism 10 (GraphPad). Outliers were detected using the ROUT method with maximum desired FDR of 1% and removed in further analyses. The Shapiro–Wilk test was used to check normality. The means were compared using unpaired Student's $t$-test; parametric $t$-test was performed on

datasets that passed the normality test, and the Mann–Whitney test was performed otherwise. For cases where the *t*-test was not applicable, the alternative statistical methods used were mentioned separately.

## Acknowledgements

We thank Dr. Ryohei Yasuda and the members of his laboratory for graciously hosting my visit and sharing their knowledge in imaging analysis. We also thank Kapil Devkota for his contributions to data analysis, Daichi Shonai, Pushpa Khanal, and Jieqing Zhao for their technical assistance in sample preparation and image acquisition, Barbara Nagy for technical assistance in EM analysis, and Ramona Rodriguiz, Christopher Means, and Ann Njoroge for conducting behavioral experiments. This work was funded by an NIH grant (R01MH126954) to SDS and an NKFIH Forefront grant (KKP #126998) to BR. Project no. RRF-2.3.1-21-2022-00001 has been implemented with the support provided by the Recovery and Resilience Facility (RRF), financed under the National Recovery Fund budget estimate, RRF-2.3.1-21 funding scheme.

## Additional information

### Funding

| Funder | Grant reference number | Author |
| --- | --- | --- |
| National Institute of Mental Health | R01MH126954 | Jaebin Kim<br>Scott H Soderling |
| National Research, Development and Innovation Office | KKP126998 | Bence Rácz |
| Recovery and Resilience Facility | RRF-2.3.1-21-2022-00001 | Bence Rácz |
| National Institutes of Health | 1S10OD028703-01 | Scott H Soderling |

The funders had no role in study design, data collection, and interpretation, or the decision to submit the work for publication.

### Author contributions

Jaebin Kim, Conceptualization, Resources, Data curation, Software, Formal analysis, Validation, Investigation, Visualization, Methodology, Writing – original draft, Project administration; Edwin Bustamante, Peter Sotonyi, Nicholas Maxwell, Methodology; Pooja Parameswaran, Software, Methodology; Julie K Kent, Resources, Data curation, Methodology, Project administration; William C Wetsel, Resources, Supervision, Methodology; Erik J Soderblom, Resources, Formal analysis, Methodology; Bence Rácz, Resources, Software, Formal analysis, Visualization, Methodology; Scott H Soderling, Conceptualization, Resources, Data curation, Supervision, Funding acquisition, Investigation, Project administration, Writing – review and editing

### Author ORCIDs

Jaebin Kim ⬡ https://orcid.org/0000-0002-1608-5868
Scott H Soderling ⬡ https://orcid.org/0000-0001-7808-197X

### Ethics

This study was conducted in accordance with National Institutes of Health guidelines and with protocols approved by the Institutional Animal Care and Use Committee (IACUC) at Duke University (protocol A167-20-08).

Reviewer #1 (Public review): https://doi.org/10.7554/eLife.97289.2.sa1
Reviewer #2 (Public review): https://doi.org/10.7554/eLife.97289.2.sa2

## Additional files

### Supplementary files
• MDAR checklist

### Data availability
All data generated by this study are included in the manuscript and supporting files. The raw proteomic data generated by this study have been deposited to the MassIVE. To view the dataset's files, log in to the MassIVE FTP server with this URL: ftp://MSV000095110@massive.ucsd.edu. Materials generated/ analyzed from this study will be provided upon request from the corresponding author.

The following dataset was generated:

| Author(s) | Year | Dataset title | Dataset URL | Database and Identifier |
| --- | --- | --- | --- | --- |
| Kim J, Soderblom EJ, Soderling SH | 2024 | Presynaptic Rac1 Interactome in vivo BioID Soderling and Kim | https://massive.ucsd.edu/ProteoSAFe/dataset.jsp?task=d509aa19025a410c88ffba12e4499f70 | MassIVE, MSV000095110 |

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
