## [Editor Report · eLife assessment]

The paper characterized a specific defect in the spatial working memory of mice with a deficit in a protein called Rac1. Rac1 inhibition was limited to the presynaptic compartment of neurons, which is significant because past work has inhibited both pre- and postsynaptic compartments. The study also identified potential effectors of Rac1. The work is **important** for these reasons, and the strength of the evidence is **exceptional**.

---

## [Referee Report · Reviewer #1 (Public review)]

- A summary of what the authors were trying to achieve:

The authors focused on Rac1, one of the most extensively studied members of the Ras superfamily of small GTPases, an intracellular signal transducer that remodels actin and phosphorylation signaling networks. They performed an extensive series of behavioral tests and found a striking result of selectively inhibiting presynaptic Rac1. Previous studies have made the claim that Rac1-mediated signaling is associated with hippocampal-dependent working memory and longer-term forms of learning and memory. Rac1 was known to modulate both pre- and postsynaptic plasticity. What was missing was selective manipulation of Rac1 function at either pre- or postsynaptic loci. Kim, Soderling, and colleagues showed that following the expression of a genetically encoded Rac1-inhibitor at presynaptic terminals, spatial working memory is selectively impaired. In contrast, Rac1 inhibition at postsynaptic sites spared the spatial working memory but affected longer-term cognitive processes.

- An account of the major strengths and weaknesses of the methods and results:

This paper is part of an ambitious research trajectory, presented in multiple rigorous studies, that combines hypothesis-free fishing for candidate signal transduction elements with precise testing of physiological and behavioral outcomes. Each of these arenas has challenges and pitfalls. This paper contains punchlines in both behavioral and cell biological areas. The effect of presynaptic Rac1 inhibition on short-term behavioral memory was convincingly demonstrated with three different behavioral tests, including a quite striking result on delayed non-matching to place task. I found the claim of a specific effect on working memory more convincing here than in previous work. On the other hand, the authors sought to clarify the presynaptic regulatory mechanisms, leveraging new advances in mass spectrometry to identify the proteomic and post-translational landscape of presynaptic Rac1 signaling. They identified particular serine/threonine kinases and phosphorylated cytoskeletal signaling and synaptic vesicle proteins that became enriched with active Rac1. They argued that phosphorylated sites in these proteins are at positions likely to have regulatory effects on synaptic vesicles. They found changes in the distribution and morphology of synaptic vesicles following presynaptic Rac1 inhibition. They also report a postsynaptic consequence, a slightly increased spine cross-sectional area.

- An appraisal of whether the authors achieved their aims, and whether the results support their conclusions:

The selective agent is the Rac1-inhibiting polypeptide W56; W56 is fused to a protein with specific subcellular localizations in neurons. Hedrick, Yasuda, et al., 2016 showed that this kind of strategy enabled a spatially targeted inhibitory effect. Collaborating with Yasuda, O'Neil in Soderling's group previously reported that Rac1 negatively regulates synaptic vesicle replenishment at both excitatory and inhibitory synapses.

In the current study by Kim et al., the goal is to interfere with Rac1 function in vivo. Once again, as in O'Neil, the functional intervention was to virally express a W56 peptide, fused to synapsin, a protein with specific subcellular localization-in this case presynaptic. The key control was to compare the effect of W56 with a scrambled sequence (Scr) in the negative control group. As verification of presynaptic efficacy, Kim found that W56-pre makes vesicles larger and further from the active zone without changing overall bouton morphology. Fresh fishing with MassSpec suggests that presynaptic vesicle proteins are affected.

I am convinced that the presynaptic Rac1 function was successfully tweaked and that this had an effect on working memory tested with 5 s intertrial intervals, in a time range where the field is hard-pressed to find robust cell biological mechanisms for memory storage. (Ion channel dynamics are an alternative, but the focus here was on cytoskeletal, not plasma membrane proteins). What was missing was a direct index of vesicle dynamics or an explanation of why a hypothetical alteration in vesicle dynamics shows up as a change in vesicle size or location. The summarizing scheme is necessarily vague; it lacks specific details about how the effect on working memory occurs, or whether it involves excitatory as opposed to inhibitory nerve terminals.

- A discussion of the likely impact of the work on the field, and the utility of the methods and data to the community:

This study reveals a previously unrecognized presynaptic role of Rac1 signaling in cognitive processes and provides insights into its potential regulatory mechanisms.

An outside observer might appreciate evidence that clearly shows that pivotal cytoskeletal cell biology is not the exclusive monopoly of either side of the synaptic cleft.

- Any additional context you think would help readers interpret or understand the significance of the work:

--Overall, it shows off the art of combining fishing with causal experiments, parallel to Steve Marx's work on L-type calcium channel modulation (Nature).

--Multiple mutations associated with human neurodevelopmental and psychiatric disorders involve genes that encode regulators of the synaptic cytoskeleton. A major, unresolved question is how the disruption of specific actin filament structures leads to the onset and progression of complex synaptic and behavioral phenotypes.

--The formation of long actin filaments along the axon's longitudinal axis is relevant to the sharing of synaptic vesicles amongst multiple boutons in so-called vesicle superpools (Chenouard & Tsien, NatComm)

---

## [Referee Report · Reviewer #2 (Public review)]

Summary:

The paper described a behavioural characterisation of mice with presynaptically-inhibited Rac1 in the hippocampus. This is followed by a BioID and phosphoproteomic analysis of Rac1, highlighting potential downstream effectors of active or non-active Rac1 and potential downstream phosphorylated targets.

Strengths:

An original molecular approach that has been established in a previous paper by the authors (PMID 34269176) to block Rac1 function exclusively at the presynapse is now utilised to characterise a link between presynaptic dysfunction and mouse behavior. The experiments and the data well-support the conclusion that the function of Rac1 has distinct outcomes on mouse behavior, depending on its site of action.

Weaknesses:

A main limitation of the study is that it lacks physiological and biochemical analysis to follow up on hits identified in a BioID and phosphoprotemic analysis of presynaptic active and non-active Rac1 variants.